# Continuity and Isolation Lead to Doubts or Dilemmas in Large Language Models

**Hector Pasten**
Faculty of Mathematics
Pontifical Catholic University of Chile & CENIA
hector.pasten@uc.cl

**Felipe Urrutia**
University of Chile & CENIA
furrutia@dim.uchile.cl

**Hector Jimenez**
University of Chile & CENIA
hjimenez@dcc.uchile.cl

**Cristian B. Calderon**
CENIA
cristian.buc@cenia.cl

**Cristóbal Rojas**
Institute for Mathematical and Computational Engineering
Pontifical Catholic University of Chile & CENIA
luis.rojas@uc.cl

**Alexander Kozachinskiy**
CENIA
alexander.kozachinskiy@cenia.cl

## Abstract

Understanding how Transformers work and how they process information is key to the theoretical and empirical advancement of these machines. In this work, we demonstrate the existence of two phenomena in Transformers, namely *isolation* and *continuity*. Both of these phenomena hinder Transformers to learn even simple pattern sequences. Isolation expresses that any learnable sequence must be isolated from another learnable sequence, and hence some sequences cannot be learned by a single Transformer at the same time. Continuity entails that an attractor basin forms around a learned sequence, such that any sequence falling in that basin will collapse towards the learned sequence. Here, we mathematically prove these phenomena emerge in all Transformers that use compact positional encoding, and design rigorous experiments, demonstrating that the theoretical limitations we shed light on occur on the practical scale.

## 1 Introduction

The massive adoption of generative artificial intelligence (AI), and in particular the use of Large Language Models (LLMs), has lead to a growing interest in understanding how these machines work and what they can and cannot compute [5]. While this endeavor has been primarily addressed from an empirical, task performance perspective [11, 9, 3], it has now become clear that a fundamental theoretical understanding of the Transformer (the architecture behind the success of LLM-based applications [19]) is a crucial step towards explaining and overcoming the observed limitations. [2, 20, 7]. This paper is a contribution to a recent trend of work devoted to the development of such an understanding by revealing the mathematical properties of Transformers [17] that underlie these limitations.

39th Conference on Neural Information Processing Systems (NeurIPS 2025).

## 1.1 Our contribution

In this work, we identify two fundamental properties of Transformers: *Isolation* and *continuity* (see below), which severely constrain their core abilities to display basic elements of intelligent behavior.

As a case study, consider the following scenario inspired by tests such as IQ. Given a sequence of symbols generated by a simple rule, an "intelligent agent" (a human or an LLM) is presented with an initial finite portion of the sequence and asked to guess the next symbol. The agent is allowed to request to see more symbols in the sequence before providing an answer. Thus, the challenge is to eventually understand the underlying pattern and apply it.

For example, if the presented sequence is "00000000000000000000", we would expect the agent to understand the underlying **constant pattern** and answer "0". If the presented sequence is now "1231231231231231231", we would still expect the agent to understand its **periodic pattern** and answer "2". A slightly more complicated sequence would be "10100100010". What is the underlying pattern? perhaps this sequence seems too short to confidently recognize it, so we request some more symbols and get

$$\text{"101001000100001000001000000"} \tag{1}$$

The pattern now is much more clear, let us call it the **increasing spacing pattern**: the 1s are separated by growing blocks of 0s, each time with one more 0. Before the last 1, we see five 0s, so we expect the agent to answer "0" to complete, after the last 1, a block with six 0s.

Given the immense practical success of LLMs, one could think that these and other sequences generated by a similarly simple rule would be easy for them to distinguish. In this work, we challenge this view and argue that this is not the case for a large class of Transformers that includes most of the modern LLMs. Namely, we show that for *decoder-only Transformers with compact positional encoding (CPE)*, **the set of sequences that they can "distinguish" in the above sense is rather limited.** As we will see below, these limitations have several strong practical implications.

We consider Transformers that, when presented with some prompt, compute a probability distribution over the set of tokens. Now, let us say that a Transformer $T$ *eventually learns* an infinite sequence of symbols $\alpha = \alpha_1 \alpha_2 \alpha_3 \ldots$ if, for any long enough prefix of $\alpha$ that is presented to $T$, when asked "what is the next symbol in the sequence?", the token that has the highest probability according to the distribution computed by $T$ is the symbol in $\alpha$ that comes after this prefix.

Here, we require that the probability of the most likely token is at least some positive constant larger than the probability of any other token. This constant may depend on $\alpha$ and can be arbitrarily small, but it has to be independent of the length of the prompt. This is exactly when the top probability can be made arbitrarily close to 1 by setting the temperature to some small but fixed positive constant, making sure that $T$ outputs the correct prediction with high probability.

**Isolation.** We establish a phenomenon of *isolation* in the learnability landscape of any decoder-only CPE Transformer $T$. More specifically, we show that any infinite sequence $\alpha$ that is eventually learned by $T$ must necessarily be *isolated* from any other infinite sequence that is also eventually learned by $T$. This means that there is a *ball* of positive radius $\delta$ around $\alpha$ in the space of infinite sequences such that no other sequence within this ball (except for those that differ from $\alpha$ in only finitely many places) can also be eventually learned by $T$. The ball is taken with respect to the relative Hamming distance, that is, it consists of all sequences that differ from $\alpha$ in a set of positions whose asymptotic frequency is at most $\delta$. This phenomenon is illustrated in Figure 1a. In other words, a Transformer required to learn two sequences that are too close to each other will face a *dilemma* in the sense that it can only learn one of them.

**Representational collapse and continuity.** What prevents a ball around $\alpha$ from containing other sequences that are also learnable by $T$ is a strong form of *representational collapse* within the ball. Namely, for any sequence $\beta$ within the ball, the output distributions of $T$ on any two long enough prefixes of $\alpha$ and $\beta$ of the same length, will be so close that the top-probability token will be the same for both prefixes, namely the next symbol of $\alpha$. Now, if $\beta$ differs from $\alpha$ at infinitely many places, $T$ must necessarily make infinitely many mistakes in predicting $\beta$, which is therefore not eventually learned by $T$.

Such representational collapse follows from Theorem 1, our main technical result, which is *continuity* of decoder-only CPE transformers. Intuitively, continuity means that making some small modifica-

tions to a prompt cannot uncontrollably change the distribution computed by a Transformer. More precisely, as long as $T$ is a decoder-only CPE Transformer, for any $\varepsilon > 0$ there exists a threshold $\delta > 0$ such that, to produce a change by more than $\varepsilon$ in the distribution computed by $T$ on a given prompt, it is necessary to change at least a $\delta$-fraction of the tokens in the prompt (excluding the last token, which we always assume remains unchanged). Notably, the value of $\delta$ for a given $\varepsilon$ depends only on $T$ and not on the length of the prompt. See Figure 1b for an illustration. In a sense, it means that the only way out from isolation is for a Transformer to give up certainty and thus provide its predictions with *doubts*. We note that Theorem 1 is actually formulated for finite sequences. The conclusions we derive from it for infinite sequences are meant to illustrate a concrete limitation to perform inductive reasoning, understood as the ability to derive general rules and principles from the presented information. The limitation for infinite sequences means that the transformer will never be able to understand the rule that generates the sequence, regardless of how long the prompt is.

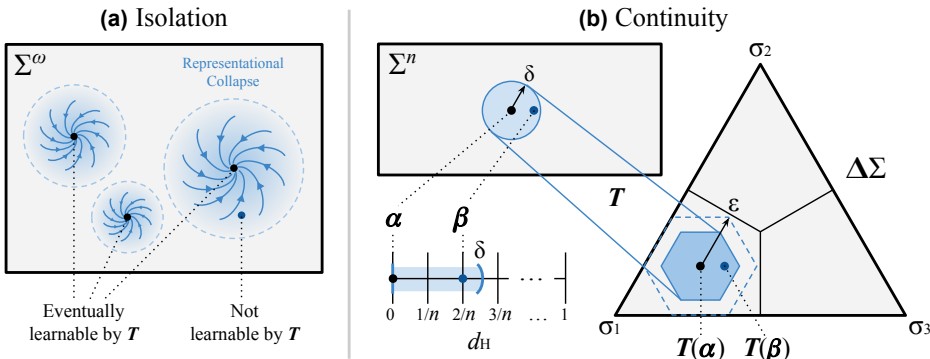

Figure 1: **Isolation and Continuity in decoder-only Transformers.** **(a) Isolation:** Illustration of the isolation phenomenon in the space of infinite sequences $\Sigma^\omega$. Only a few sequences (**black** dots) are eventually learnable by the Transformer $T$, and each is surrounded by a region where no other distinct sequence is learnable (**blue** dot). **(b) Continuity:** This figure illustrates how two similar input sequences $\alpha, \beta \in \Sigma^n$ of length $n \geq 3$ over the alphabet $\Sigma = \{\sigma_1, \sigma_2, \sigma_3\}$, which share the final token ($\alpha_n = \beta_n$) and differ in two positions ($d_H(\alpha, \beta) = 2/n \leq \delta$), are mapped by the Transformer $T$ to probability distributions with at most $\epsilon$ distance in the simplex $\Delta(\Sigma)$.

**Implications.** It is not hard to see that for any particular infinite sequence $\alpha$, there exists a decoder-only CPE Transformer that eventually learns it (one can store $\alpha$ inside the positional encoding). Arguably, a Transformer tailored like this to learn a single sequence is not very useful. We show that learning even a single additional sequence might already be problematic for LLMs. For example, isolation implies that there is no decoder-only CPE Transformer that can eventually learn both the all-0 sequence and the increasing spacing pattern sequence illustrated in equation (1). Indeed, although they differ at infinitely many positions, the frequency of positions where they differ converges to 0. This means that any positive-radius ball around the all-0 sequence must also contain the increasing spacing sequence from (1). Hence, by isolation, no decoder-only CPE Transformer $T$ can eventually lean both sequences. As a result, we see that not every pair of infinite sequences $\alpha$ and $\beta$ can be eventually learned by a single decoder-only CPE Transformer.

One may argue that the pattern in (1) is too complicated, since it essentially boils down to counting, a task in which Transformers have been reported to struggle before [1, 21]. But what about periodic sequences? They are arguably the simplest infinite family of patterns. In fact, one can show that for any *finite* set of periodic sequences, there exists a decoder-only CPE Transformer $T$ that eventually learns them all. However, our results imply that the set of *all periodic sequences* is impossible to be eventually learned by any single Transformer. Indeed, for any $T$ that learns the all-0 sequence there is, by isolation, a ball of some positive radius $\delta$ around the all-0 sequence such that no other sequence in this ball that contains infinitely many 1's can also be eventually learned by $T$. But for any $k > 1/\delta$, this ball includes the following periodic sequence:

$$\beta = \underbrace{00\ldots01}_{k}\underbrace{00\ldots01}_{k}\underbrace{00\ldots01}_{k}\ldots, \tag{2}$$

which therefore cannot be learned by $T$.

More generally, let us consider the concept of *overwhelming strings* introduced in [16]. For a given transformer $T$, a string of tokens $s$, a fixed final token $q$, and an integer $m$, we say that $T$ is "overwhelmed" by $s$ if the output of $T$ evaluated on $s$ plus any additional string $t$ and fixed final token $q$, $T(s + t + q)$, is the same regardless of the chosen string $t$, as long as its length is at most $m$. In other words, being overwhelmed by some string $s$ means that $T$ is completely insensitive to the part of the prompt that contains $t$. As pointed out by [16], this has several important practical consequences. For instance, it can be used to show "no-go" results in prompt engineering, or to prove severe limitations of transformers to compute highly sensitive functions (such as PARITY). As an immediate consequence of our work, it follows that if $\alpha$ is an infinite sequence eventually learned by $T$, then $T$ must be overwhelmed by any sufficiently long prefix $s$ of $\alpha$. Indeed, by isolation, there exists $\delta > 0$ such that appending $m$ arbitrary tokens to $s$ (but keeping the same final token) leaves the output of $T$ unchanged whenever $m/(|s| + m) < \delta$, where $|s|$ denotes the length of $s$. In particular, $T$ will be overwhelmed by any such $s$.

## 1.2 Related work

As we have already mentioned, our work shows that every Transformer is overwhelmed by infinitely many strings, a concept introduced in [16]. Their results are however of a different nature, as they focus on developing algorithms to rigorously detect whether a given Transformer $T$ is overwhelmed by a given string $s$. Our work is of theoretical nature, proving continuity and isolation and stipulating the associated limitations in decoder-only Transformers.

Continuity has been used before to demonstrate difficulties that Transformers have in learning functions where small changes in the input leads to a substantive change in the output (e.g., PARITY) [6]. Moreover, it was shown that for such functions, as the sequence length increases, the loss landscape becomes more steep, leading to further complications in the learning process [7]. Importantly, previous continuity results have only been established for *encoder-only* Transformers, where each token attends to the whole input, not only to previous tokens as in the decoder-only case (i.e., with causal masking). Under the encoder-only assumption, due to dispersion of softmax coefficients [20], flipping the value of one token leads to a $O(1/n)$-change in other tokens [6].

The same is not verbatim true for the decoder-only architecture, and this constitutes the main difficulty of extending continuity to it. The problem is that causal masking breaks the symmetry of tokens in softmax, and earlier tokens have more influence. Flipping, say, the first token, leads to significant changes not only in itself but in the few first tokens also, given that softmax coefficients are not too dispersed for them yet. A careful argument is required to show that this effect can be controlled, and this is the main technical contribution of our paper.

Representational collapse in transformers, as far as we are aware, has been previously observed by Barbero et al. [1] only in the special case of two input sequences that are identical except that we repeat the last token in one of them (which is therefore one token larger). Moreover, their result requires two important assumptions: (i) the distance between the coordinates defining the positional encoding tends to 0 as the input sequence length increases, (ii) the absence of positional information at the level of Value matrices. Whereas the second assumption is in line with the widely used and state-of-the-art rotary positional encoding method [18], it does not allow to extrapolate the results to other positional encoding methods. The first assumption is simply absent from any standard application of LLMs.

In comparison, our results apply in much more generality. We just need the value, the attention, and the activation functions to be continuous. Therefore, they apply far beyond the standard dot-product softmax attention with activations computed by MLPs, – even the Lipschitz property, crucial for the Hahn's continuity argument [6], is not required for our proof. Besides, as mentioned before, we require compactness of the positional encoding, but in contrast to [1], we can freely use it at the level of values, not only in attention.

## 1.3 Experiments

In the remainder of the paper, we first define the continuity theorem (Section 3), followed by the isolation one (Section 4). All the proofs of our theorems and lemmas can be found in Appendix A. For each theorem we present a series of experimental results to illustrate the extent to which the limitations predicted by our theorems can be observed in practice. We do this for several of

the most recent versions of modern LLMs. We investigate continuity in three applications: the all-0 sequence, a Python code syntax error detection task, and a natural language inference task. For isolation, we evaluate how well decoder-only models can generate periodic sequences. Our results not only provide strong evidence that our theoretical findings are relevant in practice, but also offer a comprehensive picture of how the specific differences among these modern architectures affect the severity of the observed limitations. Code for our experiments can be found at ⭕ furrutiav/doubts-and-dilemmas-neurips25.

## 2 Preliminaries

By $\| \cdot \|$ in this paper we mean the $l_\infty$-norm, but all our results hold for any other norm due to the equivalence of any two norms in $\mathbb{R}^d$ up to a constant factor.

**Attention layers**  The main part of the Transformer architecture (see Figure 6) is the *attention layer*.

**Definition 1.** *A d-dimensional decoder-only attention layer is a function $L\colon (\mathbb{R}^d)^* \to (\mathbb{R}^d)^*$, given by a "positional encoding" $p\colon \mathbb{N}^2 \to \mathbb{R}^d$, a continuous "value function" $val\colon \mathbb{R}^d \to \mathbb{R}^d$, a continuous "weight function" $w\colon (\mathbb{R}^d)^3 \to (0, +\infty)$, and a continuous "activation function" $F\colon \mathbb{R}^d \times \mathbb{R}^d \to \mathbb{R}^d$.*

*Given an input sequence of vectors $\bar{x} = (x_1, \ldots, x_n) \in (\mathbb{R}^d)^n$, the layer $L$ outputs a sequence of vectors $\bar{y} = (y_1, \ldots, y_n) = L(\bar{x})$, computed as follows. First, one computes the "attention weights" and "values":*

$$w_{ij} = w(x_i, x_j, p(i,j)), \qquad i, j = 1, \ldots, n, \ i \leq j$$
$$v_j = val(x_j), \qquad j = 1, \ldots, n.$$

*then a sequence $\bar{a} = (a_1, \ldots, a_n)$ of "attention vectors" as follows:*

$$a_j = \frac{w_{1j}v_1 + w_{2j}v_2 + \ldots + w_{jj}v_j}{w_{1j} + w_{2j} + \ldots + w_{jj}}, \qquad j = 1, \ldots, n,$$

*and finally, one sets $y_j = F(a_j, x_j)$, $\qquad j = 1, \ldots, n$.*

Observe that $y_j$, the $j$-th output of $L$ on input $(x_1, \ldots, x_n)$, depends only on $x_1, \ldots, x_j$. This implies that decoder-only attention layers are "prefix-monotone" functions: if a $\bar{x}_1$ is a prefix of $\bar{x}_2$, then $L(\bar{x}_1)$ is a prefix of $L(\bar{x}_2)$.

A positional encoding $p\colon \mathbb{N}^2 \to \mathbb{R}^d$ is called *compact* if there is a compact $K \subseteq \mathbb{R}^d$ such that $p(i, j) \in K$ for all $i, j \in \mathbb{N}$.

**Transformers**  By Transformers we mean functions that maps finite words over some alphabet $\Sigma$ to probability distributions over letters of $\Sigma$. To be more in line with terminology, accepted in the study of transformers, we refer to "words" as "sequences" and to their "letters" as "tokens".

The set of probability distributions over a finite set $\Sigma$ is denoted by $\Delta(\Sigma)$.

**Definition 2.** *A $d$-dimensional $k$-layer decoder-only Transformer over a finite alphabet $\Sigma$ is a function $T\colon \Sigma^* \to \Delta(\Sigma)$, given by an input embedding $e\colon \Sigma \times \mathbb{N} \to \mathbb{R}^d$, $k$ $d$-dimensional attention layers $L_1, \ldots, L_k$, and a continuous function $P\colon \mathbb{R}^d \to \Delta(\Sigma)$.*

*On an input sequence of tokens $\alpha = \alpha_1 \ldots \alpha_n \in \Sigma^n$, the output probability distribution $T(\alpha)$ is computed as follows. First, we set*

$$x_j = e(\alpha_j, j), \qquad j = 1, \ldots, n.$$

*Then we compute the composition of attention layers $L_1, \ldots, L_k$ on the input sequence of vectors:*

$$(y_1, \ldots, y_n) = L_k \circ \ldots \circ L_1(x_1, \ldots, x_n).$$

*Finally, we set $T(w) = P(y_n)$.*

An input embedding is *compact* if for some compact $K \subseteq \mathbb{R}^d$ we have $e(\sigma, i) \in K$ for all $\sigma \in \Sigma, i \in \mathbb{N}$. Overall, we call a Transformer $T$ *compact* if it uses compact input embedding and compact positional encoding in all layers.

For notational simplicity, we assume that each layer has just 1 attention head. However, our model subsumes the case when an attention layer can have $O(1)$ attention heads as we can just compute each head in a separate layer.

# 3 Continuity

Let $d_H(\alpha, \beta)$ denote the relativized Hamming distance between two sequences of tokens $\alpha, \beta \in \Sigma^n$:

$$d_H(\alpha, \beta) = \frac{|\{i \in \{1, \ldots, n\} : \alpha_i \neq \beta_i|}{n}$$

**Theorem 1.** *Let $T$ be a compact decoder-only Transformer. Then for any $\varepsilon > 0$ there exists $\delta > 0$ such that for any $n \in \mathbb{N}$, for any sequence of tokens $\alpha, \beta \in \Sigma^n$ with the same last token, if $d_H(\alpha, \beta) \leq \delta$, then $\|T(\alpha) - T(\beta)\| \leq \varepsilon$.*

**Corollary 1** (Next-token propagation principle, informal)**.** *Given $T$ a CPE decoder-only Transformer, if two prompts are very similar, end in the same token, and the next token prediction for one of them is computed with certainty, then one can expect that the next-token prediction for the other sequence is the same as for the first one.*

## 3.1 Empirical support for continuity: Zero fundamental sequence

We investigate the behavior of decoder-only language models when presented with two highly similar prompts (whose Hamming distance is small), denoted $\alpha$ and $\beta$. According to Corollary 1, if the Hamming distance is small enough, then we expect the model to produce the same next-token prediction for both sequences. In order to test this (see more details in the Appendix B.2), we define an input prompt $\alpha$ as a sequence of 190 consecutive zeros[1]. We also generate 100 sequences $\beta_\gamma^1, \beta_\gamma^2, \ldots, \beta_\gamma^{100}$ independently, where $\beta_\gamma^i$ is generated perturbing $\alpha$ at $\max(1, \lfloor \gamma \cdot 189 \rfloor)$ positions chosen uniformly at random (ignoring the last position) and $\gamma \in (0, 1/2]$ controls the proportion of differing positions. Thus, the relative Hamming distance between $\alpha$ and $\beta_\gamma^i$ is (almost) $\gamma$. All the sequences (including $\alpha$) are appended to the common instruction prefix: "`Complete the sequence with 0s and 1s:`", and submitted to the model as input. We then generate the next token $N(\alpha)$ and $N(\beta_\gamma^i)$ for every $i = 1, \ldots, 100$. We measure the model sensitivity, counting how many $\beta$'s produce the next token different from the next token of $\alpha$ (in our case 0), i.e.

$$\mathsf{NTS}_\gamma(\alpha) = |\{i \in \{1, 2, .., 100\} : N(\beta_\gamma^i) \neq N(\alpha)\}|,$$

where a higher count indicates greater sensitivity to changes in the input.

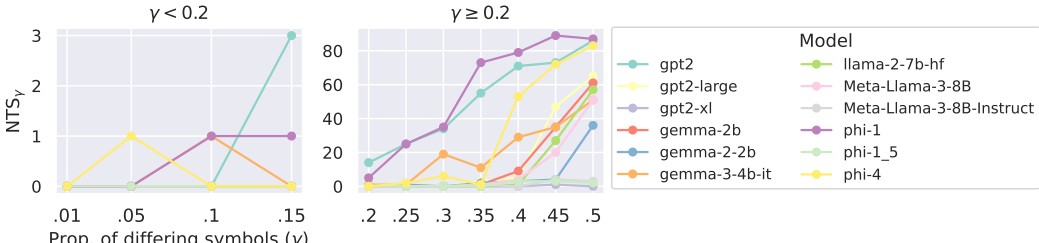

Figure 2: Sensitivity of decoder-only language models to input perturbations at $\gamma < 0.2$ and $\gamma \geq 0.2$.

As shown in Figure 2, the results support the Corollary 1. If the proportion of differing symbols is small enough ($\gamma = 0.01$, corresponding to 1 symbols), the number of diverging samples is null for all the models, suggesting the existence of an attractor (representational collapse) around the all-zero sequence. In fact, even at $\gamma = 0.05$ (9 symbols) the value of $\mathsf{NTS}_\gamma$ is 0 for all models except two, `phi-4`. Somewhat surprisingly, we observe that smaller models sometimes expose larger sensitivity (larger value of $\mathsf{NTS}_\gamma$), for example, `gpt-2` and `gpt-2-xl`.

Furthermore, we analyzed whether sensitivity is impacted by the location of the perturbations in the sequence. Across models, we found a consistent trend: sensitivity is the highest when perturbations occur at the end of the sequence (Appendix B.3). Middle positions result in lower sensitivity, and early tokens yield almost no sensitivity at all.

Additionally, we observe that modifying the standard attention (softmax) with the log-length scaled attention (*ssmax*) yields a marked increase in sensitivity (Appendix B.4). This support the assumption that compact positional encoding in attention weight posses a key role on the effect of input perturbations.

---

[1]We chose this number to ensure a small enough $\delta$ for all selected models.

## 3.2 Empirical support for continuity: Code Syntax Verification

We now turn to SYNTAXVERIFICATION (see Appendix B.5 for design explanation), a more practical LLM application. As in the previous section, we consider decoder-only language models that receive two similar prompts, $\alpha$ and $\beta$, differing up to a small Hamming distance. According to Corollary 1, for sufficiently small Hamming distances, we expect the model to produce the same next-token prediction for both prompts, even though the task requires the model to provide different outputs. Of course, different models will typically have different thresholds below which the Hamming distance classifies as being sufficiently small in this sense. However, even if some small Hamming distance is not quite below the theoretical threshold of a model, we still expect the model to predict the same next token on a large set of pairs of input sequences. SYNTAXVERIFICATION is designed precisely to visualize the extent to which this phenomenon can be observed at scales of practical relevance.

Our $\alpha$ and $\beta$ prompts share the same structure, as illustrated in Figure 10. Each sequence begins with the same main instruction (MI), followed by three *Exercises*, namely, two example exercises (shots) and one test exercise. Each *Exercise* $E_i$ in the prompt consist of four components: an *Instruction*, a *Python Code* snippet, a *Question* and an *Answer*. The $\alpha$ and $\beta$ prompts share the same first two *Exercises*, but they have a small discrepancy in the test case *Exercise*. The two prompts differ in a single token within the test *Exercise* (blue boxes in Figure 10, Appendix B.5). In the particular case of Figure 10, the $\alpha$ prompt includes the "=" token in the correct version, which is replaced by the "for" token in the $\beta$ prompt (incorrect version)[2]. For all tests, we query the LLM in the same way: "Does the following Python code compile without syntax errors? If no error is detected, return 1; otherwise, return 0.".

We built a dataset of 100 python function exercises presented in two versions, with and without syntax error. Each exercise is embedded as a final test exercise in both (correct and incorrect) formats (Figure 10). We consider a model to be *sensitive* if it produces different answers when presented with the $\alpha$ and $\beta$ prompts. More specifically, we give to the model the prompt $\alpha$ and generate a token $\sigma$. Then we compare the probability of $\sigma$ under the prompt $\alpha$ (denoted by $P(\sigma|\alpha)$) with the probability of the same token $\sigma$ under the prompt $\beta$ (denoted by $P(\sigma|\beta)$). To visualize sensitivity, we plot a point with coordinates $(P(\sigma|\alpha), P(\sigma|\beta))$. Intuitively, any substantial deviation from the diagonal line should correspond to trials where the model is sensitive, and vice versa.

Our results are depicted in Figure 3. While a vast majority of the examples do not show sensitivity, the proportion of trials where sensitivity is observed varies significantly across different models. First, we observe that model size (at least within the same family, see gemma-3 models) has an influence on sensitivity, with bigger models displaying more sensitivity. Second, phi-4, a model that is primarily trained on code, fails spectacularly, providing the same output for all cases (a rather striking result). Lastly, `Meta-llama-3-8B-Instruct` displays sensitivity in $21\%$ of the examples. These results suggest that while the theoretically required Hamming distance might be quite small and heavily depend on the particular model, the consequences of its existence may well become relevant at scales of practical interest.

## 3.3 Empirical support for continuity: Natural Language Inference

To further provide support for the continuity phenomenon at the practical scale, we evaluate an LLM on a natural language inference (NLI) task analogous to the SYNTAXVERIFICATION task. The new dataset consists of $\approx 50$ examples derived from the `babi-nli` dataset[3]. Each instance includes a premise, a hypothesis, and a label (0 or 1). To meet theoretical requirements, we lengthen the premise using neutral text from Wikipedia, creating a *long-premise* that preserves the original label and constitutes the $\alpha$ sequence. To create the corresponding $\beta$ sequence we modified a single key word at the beginning making sure that it entails a change to the original label. For example, we define premise $\alpha$ (with positive label) as `"Yann is hungry. Jason is bored. Antoine is hungry. Yann went back to the kitchen. Yann picked up the apple there..."`. We define premise $\beta$ (with the opposite label) as `"Yann is hungry. Jason is bored. Antoine is hungry. Yann`

---

[2]In practice, we define $\alpha$ as the prompt generating the greater discrepancy between the top-1 and top-2 token probability.

[3]`https://huggingface.co/datasets/tasksource/babi_nli`

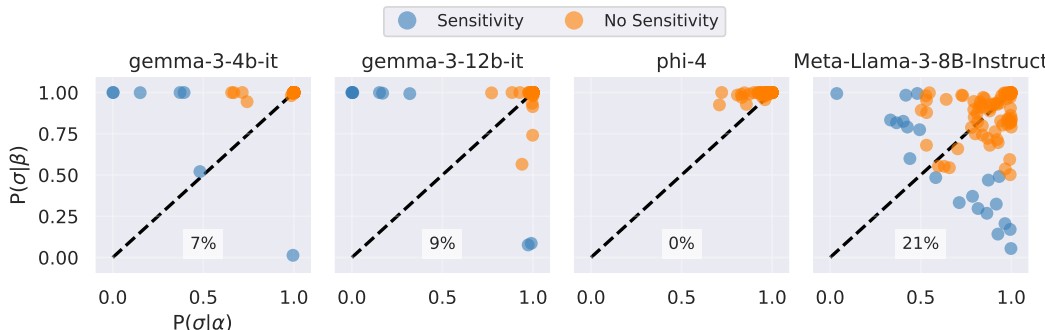

Figure 3: Sensitivity in the SYNTAXVERIFICATION task. This figure illustrates the sensitivity of five models to subtle syntactic changes in Python functions for pairs of input prompts. Each dot represents the model's probability assigned to a target token $\sigma$ under two prompts, $\alpha$ and $\beta$. **Blue** dots indicate *sensitive* cases where the model's output changed in response to the syntactic errors, as expected. **Orange** dots mark *non-sensitive* cases where the model failed to adapt its prediction, despite the change in input. Percentages in each subplot indicate the proportion of samples where the model exhibited sensitivity.

```
went back to the kitchen.  Yann picked up the pear there...". We then define the
```
hypothesis as `"Yann picked the apple because she was hungry."`.

Considering that `Meta-Llama-3-8B-Instruct` provided the best results in the SYNTAXVERIFICA-TION task, we tested this model on the new NLI task in two scenarios with different input lengths, ranging from $\approx 1000$ tokens to $\approx 7500$ tokens. As depicted in figure 4, our results show that both accuracy and sensitivity decline with input sequence length (as predicted by Corollary 1).

Our results demonstrate that, even in this simple natural language inference task, sensitivity remains fairly low. This experiment thus provides an important limitation stemming from Theorem 1 at the practical scale.

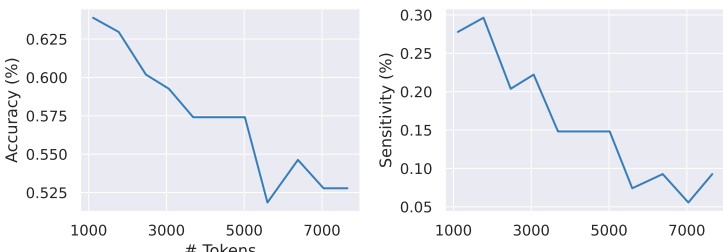

Figure 4: Accuracy and Sensitivity of `Meta-Llama-3-8B-Instruct` in the NLI task.

## 4   Isolation

We start with a formalization of the notion of eventual learnability.

**Definition 3.** *A decoder-only transformer $T$ eventually learns an infinite sequence $\alpha = \alpha_1\alpha_2\alpha_3\ldots \in \Sigma^\omega$ if there exists $\varepsilon > 0$ and $n_0 \in \mathbb{N}$ such that for all $n \geq n_0$, we have*

$$T(\alpha_1\ldots\alpha_n)(\alpha_{n+1}) \geq T(\alpha_1\ldots\alpha_n)(\sigma) + \varepsilon$$

*for any $\sigma \in \Sigma \setminus \{\alpha_{n+1}\}$.*

An easy consequence of Theorem 1 is that eventual learnability is not affected by making finitely many changes in a sequence.

**Proposition 1.** *Let $T$ be a compact decoder-only transformer, and $\alpha, \hat{\alpha} \in \Sigma^\omega$ be two infinite sequences, differing only in finitely many positions. Then $T$ eventually learns $\alpha$ if and only if it eventually learns $\hat{\alpha}$.*

We now formulate our isolation theorem. For that, we require an extension of relative Hamming distance to infinite sequences:

$$d_H(\alpha, \beta) = \liminf_{n \to \infty} d_H(\alpha_1 \ldots \alpha_n, \beta_1 \ldots \beta_n), \qquad \alpha, \beta \in \Sigma^\omega.$$

**Theorem 2.** *Let $T$ be a decoder-only compact transformer. Then for any infinite sequence $\alpha$, eventually learnable by $T$, there exists $\delta > 0$ such that no infinite sequence $\beta$ that differs from $\alpha$ in infinitely many positions and satisfies $d_H(\alpha, \beta) \leq \delta$ is eventually learnable by $T$.*

Equivalently, this theorem states that if a set of infinite sequences $S$ has a point $\alpha \in S$ such that arbitrarily close to $\alpha$ in $d_H$-distance there is a sequence from $S$ that differs from $\alpha$ at infinitely many positions, then no decoder-only compact transformer can eventually learn all sequences from $S$.

In particular, if two sequences $\alpha, \beta$ differ in infinitely many positions but satisfy $d_H(\alpha, \beta) = 0$, then no compact decoder-only transformer $T$ eventually learns both of them. For example, there is no decoder-only compact transformer that eventually learns more than one of the following sequences: (i) The all-zero sequence $(0, 0, ...)$, (ii) The indicator sequence of the powers of 2, (iii) The indicator sequence of the squares, and (iv) The indicator sequence of the primes. Indeed, we just have to note that any two of these sequences differ in infinitely many positions, but their relative Hamming distance is 0.

One can show that for any *finite* family of infinite sequences that are all a positive $d_H$-distance away one from another, there is a single decoder-only transformer $T$ that eventually learns them all. In particular, this applies to any finite subfamily of the family of periodic sequences. However, the isolation theorem implies that this is not doable for the whole family of periodic sequences.

**Corollary 2.** *There is no decoder-only compact transformer that eventually learns all periodic sequences.*

## 4.1 Empirical support for isolation: Periodic Pattern Generation

To test Corollary 2, we consider periodic sequences of the form $\beta_p^r = (0^{p-1}1)^r 0$, i.e.

$$\beta_p^r = \overbrace{0\ldots01}^{\text{1st block}}\overbrace{0\ldots01}^{\text{2nd}}\ldots\overbrace{0\ldots01}^{r\text{th}}0,$$
$$\underbrace{\phantom{0\ldots010\ldots01\ldots0\ldots01}}_{r \text{ blocks of length } p}$$

where $p$ is the period and $r$ is the number of repetitions. We construct input sequences appending $\beta_p^r$ to the common instruction prefix: "Complete the following periodic sequence with 0s and 1s:". The model is then evaluated to continue the pattern over 505-autoregressive steps.

We assess performance using two metrics: (*i*) **Success** (binary metric) captures whether the model perfectly reproduces the correct continuation. A correct sequence yields a checkmark (✓), while any deviation results in a cross (✗), and (*ii*) **Certainty** (metric between 0 and 1) is measured as the difference between the top two probabilities for the next token after $(p-2)$-autoregressive steps[4]. A larger difference indicates greater model confidence.

Figure 5 presents results from the Llama-2-7b-hf model across $r = 1, 4, 10$ repetitions and periods from 2 to 40 (see Appendix B.7 for an extended analysis to a broader set of models with varying sizes and architectures). Columns correspond to different numbers of pattern repetitions, which can be thought of varying the number of examples (here the pattern to be repeated) seen by the model prior to the generation phase. Our results show that there exists a critical period beyond which the model *cannot* successfully learn the periodic sequence correctly (first row; as predicted by Corollary 2). This critical period seems to increase with the number $r$ of examples the model is shown. Lastly, certainty (second row) shows that the difference in probability between the two top tokens displays a small dip for the next-token. This dip becomes more pronounced around the critical period, where the model initially predicts the first one token (blue dots) correctly, but then begins to generate the zero token (pink dots) instead.

---

[4]At this stage, we expect to predict the first one token following the generation of $p-2$ zeros.

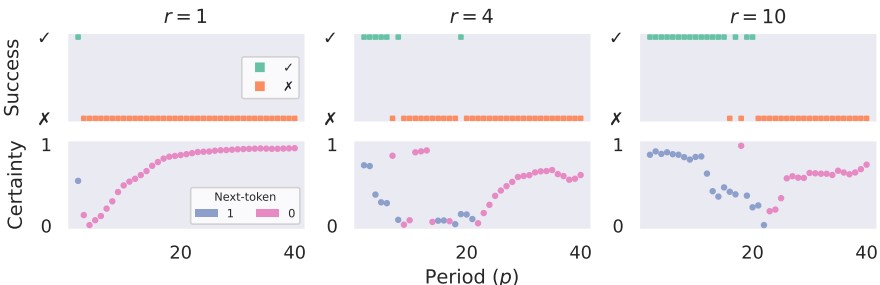

Figure 5: Evaluation of periodic sequence generation using `Llama-2-7b-hf`.

## 5   Conclusion – Doubts and Dilemmas

Our results show that Transformers (or, maybe, their developers) face inevitable dilemmas: even among very simple sequences they have to choose some that they will not be able to learn. This relies on fundamental properties of continuity and isolation but also on our assumption about the absence of *doubts* – the next token has to be predicted with certainty, its probability has to be the largest one with some margin. One could avoid dilemmas by giving up the no-doubts assumption, but one cannot avoid both – continuity and isolation imply that either dilemmas or doubts (or both) will be faced.

It is worth noting that the limitations we identify are not inherent to all sequence models. As shown in [12], simple state-space models such as RNNs can recognize any regular language. Thus, Theorem 1 can be seen as establishing a formal separation between the representational abilities of RNNs and Transformer architectures. This suggests a principled reason for why augmenting Transformers with state-space components (as done in models like Jamba) may help overcome representational collapse.

**Limitations**   The key requirement for our results is compactness of positional encoding (meaning that its vectors are bounded in norm by some absolute constant). This subsumes such standard positional encodings like sinusoidal [19] and rotary [18], but not some others like absolute positional encoding or $\log n$-scaling [4] (now referred as scalable softmax [14]). Local layer norm is subsumed by our model as well because it is a continuous transformation, as long as some positive constant $\varepsilon > 0$ is added to the denominator (as standard in practice). It is worth to point out that some theoretical works have considered layer norm with $\varepsilon = 0$. In this regime, layer norm falls out of the scope of our results, as it is no longer continuous (and it is not even defined when the denominator is 0). Using layer norm with $\varepsilon = 0$, Chiang and Cholak [4] compute PARITY with arbitrarily high certainty, thus escaping continuity limitations (see also [8] where these limitations are avoided due to the use of unbounded positional encoding).

Another potential way to escape continuity limitations is the use of chain-of-thought (CoT; see Appendix B.6 for preliminary analyses). As long as just $O(1)$ CoT iterations are allowed, this is subsumed by our results (our model allows for any constant number of layers, so we could just add more of them, simulating each CoT inference in a new layer). Now, when the number of CoT iterations grows with the input length, this logic breaks, and the models might potentially become much more intelligent. Indeed, on the theory side, it was shown that transformers with unbounded number of CoT inferences become Turing complete [15, 13, 10]. However, none of these results is obtained for softmax with compact positional encoding, leaving the theoretical power of CoT-equipped transformers in this regime open.

**Acknowledgments**   All authors are funded by the National Center for Artificial Intelligence CENIA FB210017, Basal ANID. Kozachinskiy is supported by ANID Fondecyt Iniciación grant 11250060.

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

# A  Missing Proofs

## A.1  Proof of Theorem 1

*Proof.* Given two sequences of tokens $\alpha = \alpha_1 \ldots \alpha_n, \beta = \beta_1 \ldots \beta_n \in \Sigma^n$ with the same last token and $d_H(\alpha, \beta) \leq \delta$, we input them into the Transformer as sequences of vectors

$$x_1 = e(\alpha_1, 1), \ldots, x_n = e(\alpha_n, n), \qquad \widehat{x}_1 = e(\beta_1, 1), \ldots, \widehat{x}_n = e(\beta_n, n). \tag{3}$$

Since the input embedding is compact, all vectors $x_i, \widehat{x}_i$ belong to a fixed compact set $K$ (not depending on $n, u, v$). The last vectors $x_n, \overline{x}_n$ coincide (given that the last token of $\alpha$ and $\beta$ coincide). Besides that, sequences of input vectors coincide in at least $(1 - \delta)n$ positions ($\alpha$ and $\beta$ coincide in at least $(1 - \delta)n$ positions).

We have to show that the last vectors $y_n^{(t)}, \widehat{y}_n^{(t)}$ will stay sufficiently close throughout all layers of our transformer $T$, where

$$(y_1^{(t)}, \ldots, y_n^{(t)}), \qquad (\widehat{y}_1^{(t)}, \ldots, \widehat{y}_n^{(t)})$$

are output sequences after $t$ attention layers of $T$ on $x = (x_1, \ldots, x_n)$ and on $\widehat{x} = (\widehat{x}_1, \ldots, \widehat{x}_n)$, respectively. More precisely, for any $\varepsilon > 0$, we have to show that the existence of $\delta > 0$ such that conditions $d_H(u, v) \leq \delta, u_n = v_n$ imply $\|y_n^{(t)} - \widehat{y}_n^{(t)}\| \leq \varepsilon$.

We show that through induction by the number of layers. To make this work, we have to strengthen the statement we are proving by induction – it will not be enough to just show that after one attention layer, the last vectors stay sufficiently close. We will also have to maintain that sequences of vectors stay sufficiently close to each other "globally". We will establish this through a lemma, staying roughly the following: for any attention layer $L$, if two input sequences of vectors are sufficiently close globally, and their last vectors are also sufficiently close, then the output sequences stay sufficiently close globally, and the last output vector stay sufficiently close as well.

We define "global similarity" between two sequences of vectors as follows. Given $x = (x_1, \ldots, x_n) \in (\mathbb{R}^d)^n$ and $\widehat{x} = (\widehat{x}_1, \ldots, \widehat{x}_n) \in (\mathbb{R}^d)^n$, define $sim(x, \widehat{x})$ as the minimal $\delta \geq 0$ such that $\|x_i - \widehat{x}_i\| \leq \delta$ for at least $(1 - \delta)n$ positions $i \in \{1, \ldots, n\}$.

Observe that for input sequences in (3), we have $sim(x, \widehat{x}) \leq \delta$ (just because they coincide in at least $(1 - \delta)n$ positions). The following lemma finishes the proof of the theorem, establishing that global similarity + last-position similarity is preserved through an attention layer.

**Lemma 1.** *Let $L$ be a decoder-only attention layer with compact positional encoding, and let $K \subseteq \mathbb{R}^d$ be a compact set. Then for any $\varepsilon > 0$ there exists $\delta > 0$ such that the following holds. For any $n \in \mathbb{N}$, and for any two sequences of vectors $x, \widehat{x} \in K^n$, we have:*

$$sim(x, \widehat{x}) \leq \delta, \ \|x_n - \widehat{x}_n\| \leq \delta \implies sim(y, \widehat{y}) \leq \varepsilon, \ \|y_n - \widehat{y}_n\| \leq \varepsilon,$$

*where*

$$(y_1, \ldots, y_n) = L(x), \qquad (\widehat{y}_1, \ldots, \widehat{y}_n) = L(\widehat{x}).$$

The only thing it remains to note, besides the proof of Lemma 1, is why after any number of layers, the vectors $y_i^{(t)}, \widehat{y}_i^{(t)}$ belong to some compact set $K$, depending solely on the transformer but not on $n, u, v$. This is proved by induction over layers by continuity of the value and activation functions $val$ and $F$. The output in the $n$-th position of an attention layer is computed as $y_n = F(a_n, x_n)$, where $x_n$ is the input vector to the attention layer in the $n$-th position, and $a_n$ is the attention vector in that position. All vectors $x_n$ come from a compact $K$. Value vectors $v_n = val(x_n)$ thus belong to $val(K)$, which is a compact set by continuity of $val$. Take now any closed ball $B$ containing both $K$ and $val(K)$. Vectors $a_n$, as convex combinations of $v_n$'s, belong to $B$. It remains to observe that $F(B \times B)$ is compact by continuity of $F$.

### A.1.1  Proof of Lemma 1

It turns out that it is enough to establish the following weaker version of Lemma 1, where we forget about global similarity of output sequences $y, \widehat{y}$.

**Lemma 2.** *Let $L$ be a decoder-only attention layer with compact positional encoding, and let $K \subseteq \mathbb{R}^d$ be a compact set. Then for any $\varepsilon > 0$ there exists $\delta > 0$ such that the following holds. For any $n \in \mathbb{N}$, and for any two sequences of vectors $x, \widehat{x} \in K^n$, we have:*

$$sim(x, \widehat{x}) \le \delta, \ \|x_n - \widehat{x}_n\| \le \delta \implies \|y_n - \widehat{y}_n\| \le \varepsilon,$$

*where*

$$y = (y_1, \ldots, y_n) = L(x), \qquad \widehat{y} = (\widehat{y}_1, \ldots, \widehat{y}_n) = L(\widehat{x}).$$

Let us start by deriving Lemma 1 from Lemma 2. We have to derive that if $\delta$ is small enough, then not only the last two output vectors $y_n, \widehat{y}_n$ are $\varepsilon$-close (as Lemma 2 says), but at least $(1 - \varepsilon)n$ output vectors are $\varepsilon$-close.

Imagine we start with two sequences of vectors $x, \widehat{x} \in K^n$ such that $sim(x, \widehat{x}) \le \delta$ and $\|x_n - \widehat{x}_n\| \le \delta$. Let $E \subseteq \{1, \ldots, n\}$ be the set of "bad positions" (where $\|x_j - \widehat{x}_j\| > \delta$). By definition of the similarity distance, its "relative size" (the fraction $|E|/n$) is at most $\delta$. Next, for any $\delta_1 > 0$, by choosing $\delta$ to be small enough, we can achieve that the relative size of $E$ in the restriction to the first $j$ positions, for any $j \ge \varepsilon n/2$, does not exceed $\delta_1$. For instance, by setting $\delta = (\delta_1 \varepsilon)/2$, we can bound:

$$\frac{|E \cap \{1, \ldots, j\}|}{j} \le \frac{|E|}{j} \le \frac{\delta n}{\varepsilon n/2} = \delta_1 n.$$

In particular, we can do that for $\delta_1$ with which the conclusion of Lemma 2 is true for $\varepsilon$. As we are free to choose $\delta$, we can also assume that $\delta \le \delta_1$. We claim now that $\|y_j - \widehat{y}_j\| \le \varepsilon$ for any $j \ge \varepsilon n/2, j \notin E$. This is because all input vectors in positions not from $E$ are $\delta_1$-close (because they are even $\delta$-close), this includes $x_j, \widehat{x}_j$ as $j \notin E$, and the number of the first $j$ positions not from $E$ is at least $(1 - \delta_1)j$.

As a result, we can have $\|y_j - \widehat{y}_j\| > \varepsilon$ only for $j \in E$ or for $j < \varepsilon n/2$. Thus, we can bound the number of such positions by $\varepsilon n/2 + \delta n$. By making sure that $\delta < \varepsilon/2$, we obtain that $sim(y, \widehat{y}) \le \varepsilon$.

It remains to establish Lemma 2.

*Proof of Lemma 2.* Take a closed ball $B$ with the center at $0$ that contains $K, val(K)$, and $p(i, j)$ for all $i, j \in \mathbb{N}$. Such $B$ exists because $K$ is compact, $val$ is continuous which means that $val(K)$ is compact, and because the positional encoding is compact meaning that $p(i, j)$ all belong to some fixed compact for $i, j \in \mathbb{N}$. Observe that on both inputs, $a_j$'s (attention vectors), also belong to $B$ as convex combinations of values vectors. Let $R$ be the radius of $B$.

We have

$$y_n = F(a_n, x_n), \qquad \widehat{y}_n = F(\widehat{a}_n, \widehat{x}_n),$$

where $F$ is the activation function of $L$, and $a_n, \widehat{a}_n$ are the $n$-th attention vectors on inputs $x$ and $\widehat{x}$, respectively. By uniform continuity of $F$ on the compact $B \times B$, there exists $\delta_1 > 0$ such that

$$\|a_n - \widehat{a}_n\| \le \delta_1, \ \|x_n - \widehat{x}_n\| \le \delta_1 \implies \|F(a_n, x_n) - F(\widehat{a}_n, \widehat{x}_n)\| \le \varepsilon.$$

It now suffices to show the existence of $0 < \delta \le \delta_1$ such that

$$sim(x, \widehat{x}) \le \delta, \ \|x_n - \widehat{x}_n\| \le \delta \implies \|a_n - \widehat{a}_n\| \le \delta_1.$$

Let $w : (\mathbb{R}^d)^3 \to (0, +\infty)$ be the weight function of $L$. We apply it only to inputs from a fixed compact set $B^3$. Hence, we can assume that for some universal constants $0 < c < C$, the function $w$ takes values in $[c, C]$. Moreover, by the uniform continuity of $w$ on $B^3$, if we change each of 3 inputs by at most $\delta$ in the norm, the output value changes by at most $c_\delta$, for some $c_\delta \to 0$ as $\delta \to 0$.

Similarly, there exists $d_\delta$ with $d_\delta \to 0$ as $\delta \to 0$ such that $\|val(x) - val(\widehat{x})\| \le d_\delta$ for all $x, \widehat{x} \in K$ with $\|x - \widehat{x}\| \le \delta$.

The norm of the difference $a_n - \widehat{a}_n$ can be bounded as:

$$\|a_n - \widehat{a}_n\| = \left\| \frac{\sum_{i=1}^{n} (w_{in} v_i \widehat{W}_n - \widehat{w}_{in} \widehat{v}_i W_n)}{W_n \widehat{W}_n} \right\| \le \frac{\sum_{i=1}^{n} \left\| w_{in} x_i \widehat{W}_n - \widehat{w}_{in} \widehat{v}_i W_n \right\|}{W_n \widehat{W}_n}, \qquad (4)$$

where

$$v_i = val(x_i), \qquad \widehat{v}_i = val(\widehat{x}_i),$$
$$w_{in} = w(x_i, x_n, p(i, n)), \qquad \widehat{w}_{in} = w(\widehat{x}_i, \widehat{x}_n, p(i, n)), \qquad i = 1, \ldots, n,$$
$$W_n = w_{1n} + \ldots + w_{nn}, \qquad \widehat{W}_n = \widehat{w}_{1n} + \ldots + \widehat{w}_{nn}.$$

Let $E$ denote the set of positions $i \in \{1, \ldots, n\}$ with $\|x_i - \widehat{x}_i\| > \delta$. Since $sim(x, \widehat{x}) \leq \delta$, we have $|E| \leq \delta n$. Since we are additionally given that $\|x_n - \widehat{x}_n\| \leq \delta$, by definition of $c_\delta$, we have:

$$|w_{in} - \widehat{w}_{in}| \leq c_\delta \qquad \text{for } i \notin E. \tag{5}$$

In turn, by definition of $d_\delta$, we have:

$$\|v_i - \widehat{v}_i\| \leq d_\delta \qquad \text{for } i \notin E. \tag{6}$$

Now, since the function $w$ takes values in the interval $[c, C]$, we obtain the following bound:

$$|w_{in} - \widehat{w}_{in}| \leq 2C \qquad \text{for } i \in E. \tag{7}$$

From (5–7), using the bound $|E| \leq \delta n$, we derive:

$$|W_n - \widehat{W}_n| \leq |w_{1n} - \widehat{w}_{1n}| + \ldots + |w_{nn} - \widehat{w}_{1n}| \leq c_\delta n + 2C|E| \leq (c_\delta + 2C\delta)n \tag{8}$$

(importantly, the coefficient before $n$ in the last upper bound goes to 0 as $\delta \to 0$).

We now upper bound the right-hand side of (4). First, the denominator there is at least $c^2 n^2$, because the weight function is at least $c$ for inputs under consideration. It remains to upper bound the numerator by an expression $f(\delta)n^2$ for some function $f(\delta) \to 0$ as $\delta \to 0$. Each term in the numerator we bound using the triangle inequality:

$$\|w_{in}\widehat{W}_n v_i - \widehat{w}_{in} W_n \widehat{v}_i\| \leq \|(w_{in} - \widehat{w}_{in})\widehat{W}_n v_i\|$$
$$+ \|\widehat{w}_{in}(\widehat{W}_n - W_n)v_i\|$$
$$+ \|\widehat{w}_{in} W_n (v_i - \widehat{v}_i)\|.$$

We have $w_{in}, \widehat{w}_{in} \leq C$, $W_n, \widehat{W}_n \leq Cn$, and $\|v_i\|, \|\widehat{v}_i\| \leq R$ (the weight function is bounded from above by $C$, and value vectors $v_i, \widehat{v}_i$ come from $B$, the ball of radius $R$ with the center at the origin). Overall, we get:

$$\|w_{in}\widehat{W}_n x_i - \widehat{w}_{in} W_n \widehat{x}_i\| \leq \left(|w_{in} - \widehat{w}_{in}| \cdot CR + |\widehat{W}_n/n - W_n/n| \cdot CR + \|v_i - \widehat{v}_i\| \cdot C^2\right)n$$

For $i \notin E$, this expression is bounded by $o(1)n$ as $\delta \to \infty$ by (5), (6), and (8). For $i \in E$, this expression is $O(n)$. Overall, the numerator in (4) is upper bounded by $o(1)n^2 + |E| \cdot O(n) \leq o(1)n^2 + O(\delta n^2) = o(1)n^2$ as $\delta \to 0$, as required. $\qquad\Box$

$\Box$

**Remark 1.** *Since for Encoders the proportion of exceptions does not change with the iterations (as in the decoder-only), the Theorem also holds for them (and the argument is actually simpler).*

### A.2 Proof of Proposition 1

We show that if a compact decoder-only transformer $T$ eventually learns $\alpha$, and $\widehat{\alpha}$ differs from $\alpha$ in finitely many places, then $T$ eventually learns $\widehat{\alpha}$ too. By definition, there exist $\varepsilon > 0$ and $n_0$ such that for all $n \geq n_0$, we have:

$$T(\alpha_1 \ldots \alpha_n)(\alpha_{n+1}) \geq T(\alpha_1 \ldots \alpha_n)(\sigma) + \varepsilon \tag{9}$$

for any $\sigma \in \Sigma \setminus \{\alpha_{n+1}\}$. We show that the same holds for $\widehat{\alpha}$ and for all large enough $n$ with $\varepsilon/2$. By Theorem 1, there exists $\delta > 0$ such that on any two sequences of tokens with the same length, the same last token, and of relative Hamming distance at most $\delta$, the output distributions of $T$ are $(\varepsilon/10)$-close in $\ell_\infty$-norm. Since $\alpha$ and $\widehat{\alpha}$ differ just in finitely many places, we have $\alpha_n = \widehat{\alpha}_n, \alpha_{n+1} = \widehat{\alpha}_{n+1}$ and $d_H(\alpha_1 \ldots \alpha_n, \widehat{\alpha}_1 \ldots \widehat{\alpha}_n) \leq \delta$ for all large enough $n$. For such $n$, if we replace every occurrence of $\alpha$ by $\widehat{\alpha}$ in (9), the left-hand and the right-hand side change by at most $\varepsilon/10$, preserving the inequality with $\varepsilon/2$, as required.

### A.3 Proof of Theorem 2

If $\alpha$ is eventually learnable by $T$, by definition, there exist $\varepsilon > 0$ and $n_0$ such that for all $n \geq n_0$, we have:

$$T(\alpha_1 \ldots \alpha_n)(\alpha_{n+1}) \geq T(\alpha_1 \ldots \alpha_n)(\sigma) + \varepsilon \tag{10}$$

for any $\sigma \in \Sigma \setminus \{\alpha_{n+1}\}$. By Theorem 1, there exists $\delta > 0$ such that for any two finite sequences of tokens that have the same length, the same last token, and are of relative Hamming distance at most $\delta$, the output distributions of $T$ on them are $\varepsilon/3$-close in the $\ell_\infty$-norm. We now take an arbitrary infinite sequence $\beta$ that differs from $\alpha$ in infinitely many places and satisfies $d_H(\alpha, \beta) \leq \widehat{\delta} = \min\{\delta/3, 1/3\}$, and show that $\beta$ is not eventually learnable by $T$.

To this end, it is enough to show that

$$T(\beta_1 \ldots \beta_n)(\alpha_{n+1}) > T(\beta_1 \ldots \beta_n)(\beta_{n+1}), \tag{11}$$

for infinitely many $n$. This would contradict eventual learnability of $\beta$ by $T$ as $\beta_{n+1}$ has to be the top-probability token of $T(\beta_1 \ldots \beta_n)$ starting from some $n$.

We take an arbitrary $N_0 \in \mathbb{N}$ and show the existence of $n \geq N_0$ for which (11) holds. Since

$$d_H(\alpha, \beta) = \liminf_{n \to \infty} d_H(\alpha_1 \ldots \alpha_n, \beta_1 \ldots \beta_n) \leq \widehat{\delta},$$

there exists $m \geq \max\{2N_0, 2n_0\}$ such that $d_H(\alpha_1 \ldots \alpha_m, \beta_1 \ldots \beta_m) \leq 1.1\widehat{\delta}$. The sequences $\alpha_1 \ldots \alpha_m$ and $\beta_1 \ldots \beta_m$ cannot differ in all positions of the second half of these sequences because their relative Hamming distance is bounded by $1.1\widehat{\delta} \leq 1.1 \cdot (1/3) < 1/2$. Hence, we have $\alpha_\ell = \beta_\ell$ for some $\ell \in [m/2, m]$. The relative Hamming distance between $\alpha_1 \ldots \alpha_\ell$ and $\beta_1 \ldots \beta_\ell$ is at most twice the relative Hamming distance between $\alpha_1 \ldots \alpha_m$ and $\beta_1 \ldots \beta_m$. Indeed, the first pair can only have fewer differences, and the length of sequences in the first pair (that goes into the denominator in the relative Hamming distance) is at most twice smaller than in the second pair. This gives us

$$d_H(\alpha_1 \ldots \alpha_\ell, \beta_1 \ldots \beta_\ell) \leq 2.2\widehat{\delta} \leq 2.2(\delta/3) < \delta$$

for some $\ell \geq m/2 \geq \max\{N_0, n_0\}$ such that $\alpha_\ell = \beta_\ell$. Take the smallest $n \geq \ell$ such that $\alpha_{n+1} \neq \beta_{n+1}$ which exists because $\alpha$ and $\beta$ have infinitely many differences. Observe that:

$$d_H(\alpha_1 \ldots \alpha_n, \beta_1 \ldots \beta_n) \leq d_H(\alpha_1 \ldots \alpha_\ell, \beta_1 \ldots \beta_\ell) \leq \delta$$

because $\alpha_1 \ldots \alpha_n$ and $\beta_1 \ldots \beta_n$ are obtained from $\alpha_1 \ldots \alpha_\ell$ and $\beta_1 \ldots \beta_\ell$ by appending some number of equal tokens. By definition of $\delta$, the distributions $T(\alpha_1 \ldots \alpha_n)$ and $T(\beta_1 \ldots \beta_n)$ are $(\varepsilon/3)$-close in the $\ell_\infty$-norm (observe that $\alpha_n = \beta_n$ as otherwise we could take smaller $n$, so the last tokens coincide and continuity can be used). Since $n \geq \ell \geq n_0$, we have (10) for $n$. If we replace $T(\alpha_1 \ldots \alpha_n)$ by $T(\beta \ldots \beta_n)$, both the left-hand and the right-hand sides change by at most $\varepsilon/3$, meaning strict inequality is preserved for any $\sigma$, in particular for $\sigma = \beta_{n+1}$. We thus obtain (11) for $n \geq \ell \geq N_0$, as required.

### A.4 Proofs of Corollary 2

For Corollary 2, assume for contradiction there is a decoder-only compact transformer $T$ that eventually learns all periodic sequences. In particular, it learns the all-0 sequence, and by Theorem 2 there exists $\delta > 0$ such that no sequences $\beta$, having infinitely many 1s and satisfying $d_H(\alpha, \beta) \leq \delta$ is eventually learnable by $T$. On the other hand, there is a periodic sequence $\beta$, satisfying these restrictions, namely

$$\beta = \underbrace{00 \ldots 01}_{k} \underbrace{00 \ldots 01}_{k} \underbrace{00 \ldots 01}_{k} \ldots$$

for any $k \geq 1/\delta$. It has infinitely many 1s, but $d_H(\alpha, \beta) = 1/k \leq \delta$, a contradiction.

## B  Supporting figures

### B.1  Visualization of Decoder-only Architecture

Figure 6 presents a high-level visualization of a standard decoder-only transformer, illustrating how an input string $\alpha$ is processed through $k$ attention layers in a causal and sequential manner. Each

token is first embedded via a function $e$, and then transformed layer by layer—respecting the prefix-monotonicity of the architecture, where the output at position $j$ depends only on the first $j$ tokens. This structure enforces the autoregressive property fundamental to decoder-only transformers. The final layer produces a vector $y_n$, which is mapped to a probability distribution over the vocabulary via a projection function $P$. This figure serves to clarify the computational assumptions underlying our theoretical results.

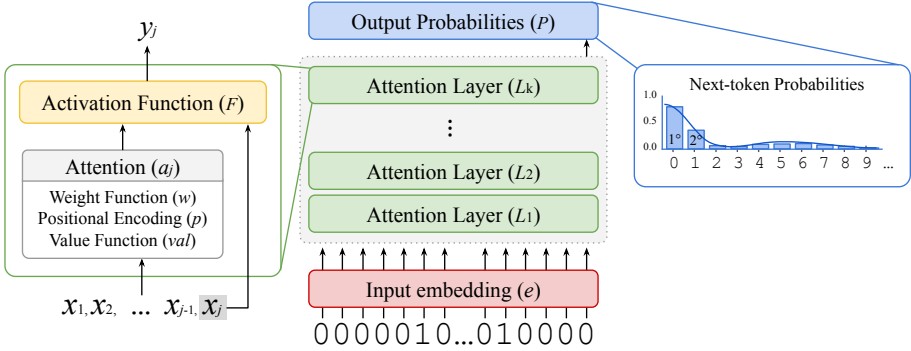

Figure 6: Schematic of a decoder-only transformer.

## B.2 Visualization of Continuity

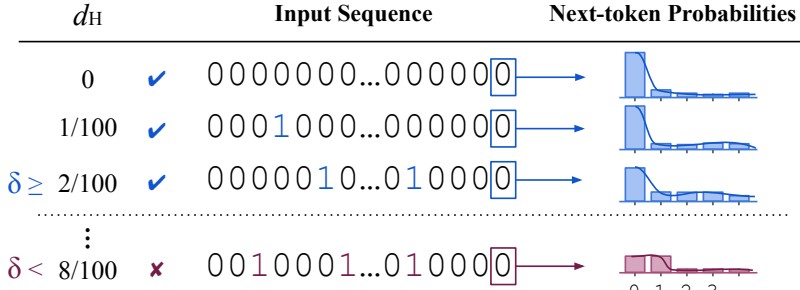

Figure 7: Illustration of continuity in decoder-only transformers under small input perturbations.

Figure 7 visualizes the continuity property established in Corollary 1, which shows that decoder-only transformers are stable under small input perturbations. In this example, we compare two sequences: $\alpha$, consisting of 100 zeros, and $\beta$, which is derived from $\alpha$ by flipping some zeros to ones. The relative Hamming distance $d_H(\alpha, \beta)$ determines whether the perturbation is within a predefined threshold $\delta$. When this threshold is satisfied, the transformer's next-token output distribution remains within a small $\varepsilon$ distance of the original. The figure highlights both perturbed sequences that respect the $\delta$-constraint and those that do not, illustrating how small input changes can result in correspondingly small or large changes in the output distribution.

## B.3 Effect of Divergent Positions

We explore how the sensitivity of models varies with the position of input perturbations. Specifically, we measure the behavior of the model when a fraction of the symbols in a sequence of zeros are flipped to ones. To parametrically control where these flips occur along the input sequence, we sample discrete positions using the Beta-Binomial distribution, whose probability mass function is given by

$$\text{BETABINOMIAL}(k \mid n, u, v) = \binom{n}{k} \frac{B(k + u, n - k + v)}{B(u, v)},$$

where $k \in \{0, 1, \ldots, n\}$ denotes the position index, $n$ is the sequence length (equal to $189$ in our experiments), $B(\cdot, \cdot)$ denotes the *Beta function*, and $u$, $v$ are the shape parameters. These parameters control the positional bias of the perturbations: toward the beginning ($u \ll v$), center ($u = v$, with $u, v \geq 1$), or end ($u \gg v$) of the sequence.

The plots in Figure 8 show the next-token sensitivity $\mathsf{NTS}_\gamma$ over 8 different positional biases of the perturbed tokens. We observe a clear trend across models: perturbations near the end of the input have a significantly greater impact on the model's output compared to those near the beginning or middle. This reflects a position-dependent sensitivity, where later tokens carry more influence over the model's immediate prediction.

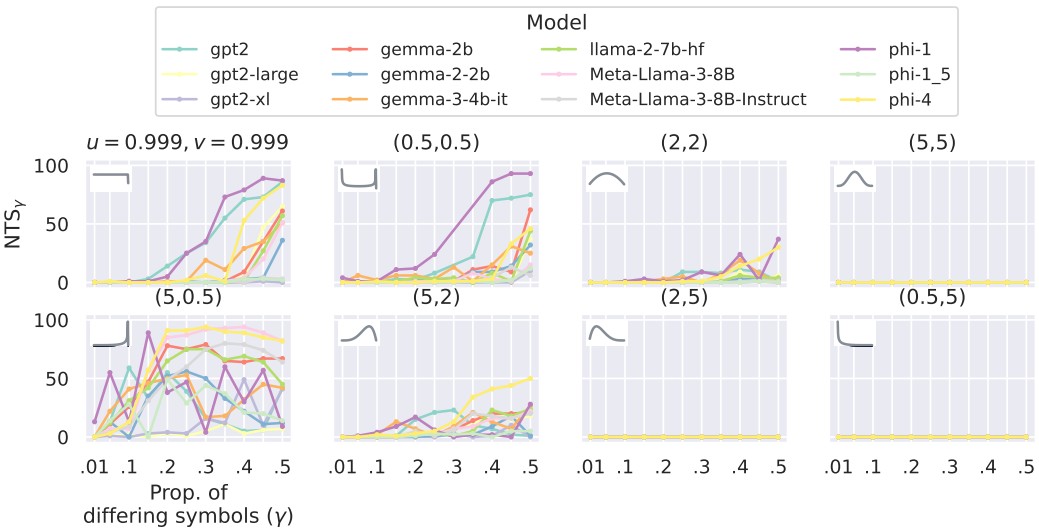

Figure 8: Sensitivity of decoder-only language models to input perturbations, visualized across different Beta-Binomial settings. The corresponding shape parameters are provided in the title and the probability mass function is displayed in the upper left corner of each panel.

## B.4  Boosting sensitivity

We investigate how modifications to the attention aggregation function affect a model's sensitivity to small input perturbations in the setup described in Section 3.1. In particular, we compare the standard *softmax* attention with *ssmax*[5], a variant designed to increase sensitivity. Originally introduced as *log-length scaled attention* by Chiang and Cholak [4] and later revisited as *scalable softmax* by Nakanishi [14], this formulation amplifies differences in logits based on sequence length. As illustrated in Figure 9, replacing softmax with ssmax significantly increases next-token variability under small input changes.

## B.5  Visualization of Code Syntax Verification

Figure 10 illustrates the SYNTAXVERIFICATION task, where the model is presented with a sequence of Python function snippets and asked to determine whether the final snippet is syntactically correct. Each prompt consists of two small examples ($F_1$ and $F_2$) with correctness annotations, followed by a third large example ($F_3$), which is the target for prediction. We construct two versions of each prompt—one where $F_3$ is correct and one where it contains a subtle syntax error (e.g., incorrect use of a keyword such as `for`). These two prompts differ by only a few tokens, allowing us to evaluate whether the model is sensitive to small but meaningful syntactic changes.

We extend our analysis for SYNTAXVERIFICATION task for multiple models versions (see Figure 11). As expected, instruct-tuned models perform significantly better on this task than base models,

---

[5]$\mathsf{SSMAX}_s(z) = \mathsf{SOFTMAX}((s \log n)\, z)$, where $z$ denotes logits of length $n$, and $s \in (0, 1]$ is a scaling factor, defaulting to 1

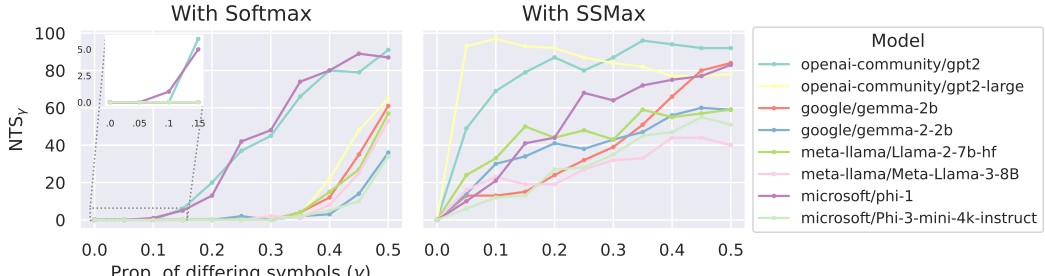

Figure 9: **Next-token sensitivity** of decoder-only language models to input perturbations. Each line shows the number of sample sequences (out of 100) that produce a different next-token than zero, as a function of the proportion of differing symbols. **Left:** Models with standard softmax in attention layers. **Right:** Models replaced with ssmax in attention layers.

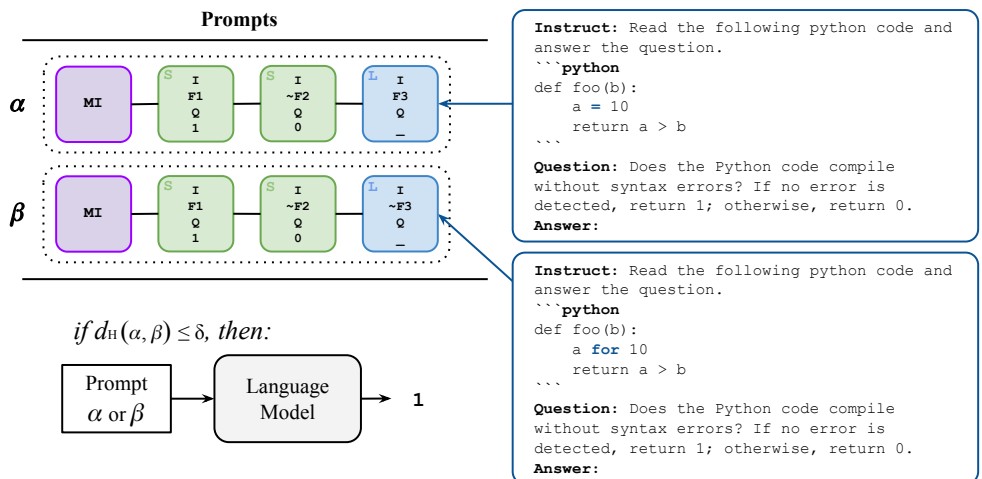

Figure 10: Visualization of the SYNTAX VERIFICATION task. The model is prompted with Python code snippets and must predict whether the final function contains a syntax error. MI refers to the main instruction in the prompt: *You are a Python expert. Read the following instructions carefully and respond to the questions.*

consistent with their enhanced ability to follow instructions and answer questions. Surprisingly, even for relative simple prompts [6] error rates remain high. Instruct models still display levels of errors above 15%, while base models above 80%.

## B.6 Syntax Code Verification on Reasoning Models

We extend our analysis for SYNTAX VERIFICATION task for reasoning models `o3-mini` and `o4-mini`, trained with long instances of CoT (Figure 12). Unsurprisingly, we observe that these models are much better at solving the task. Strikingly, we observe that for such simple task prompts, these powerful models still display levels of errors greater or equal to 20%. These result suggest that CoT cannot always escape continuity, which continues to affect the performance at scales of practical relevance, even for advanced reasoning models.

---

[6]Here we consider a smaller function dataset compared to standard dataset used for models in Section 3.2 and reasoning models in Section B.6

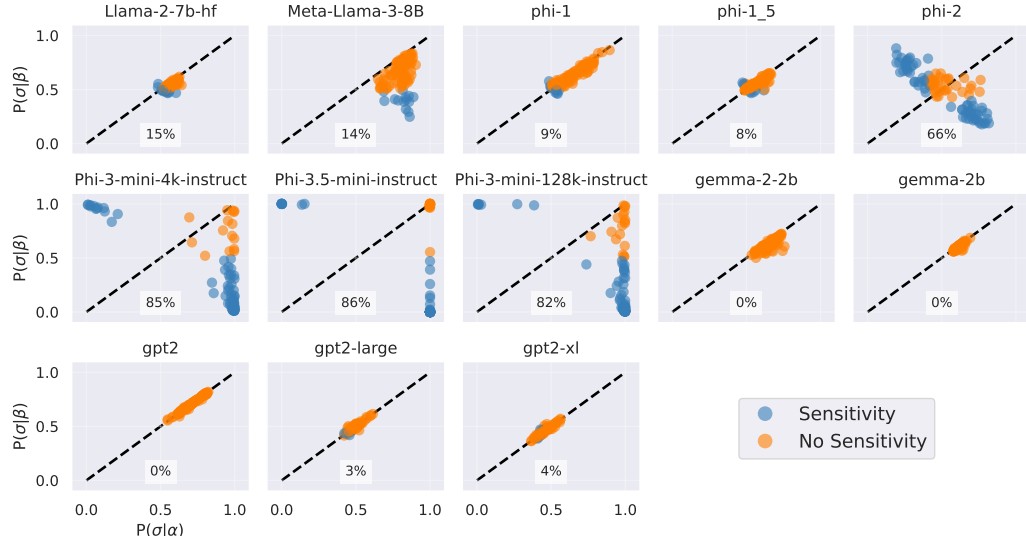

Figure 11: Sensitivity in the SYNTAXVERIFICATION task. This figure illustrates the sensitivity of five models to subtle syntactic changes in Python functions for pairs of input prompts. Each dot represents the model's probability assigned to a target token $\sigma$ under two prompts, $\alpha$ and $\beta$. **Blue** dots indicate *sensitive* cases where the model's output changed in response to the syntactic errors, as expected. **Orange** dots mark *non-sensitive* cases where the model failed to adapt its prediction, despite the change in input. Percentages in each subplot indicate the proportion of samples where the model exhibited sensitivity. For this analysis, we consider a simpler function dataset (i.e functions with less than 100 tokens) in contrast to the main experiment, which involves longer functions (over 100 tokens) for `gemma-3-4b-it`, `gemma-3-12b-it`, `phi-4` and `Meta-Llama-3-8B-Instruct` and reasoning models.

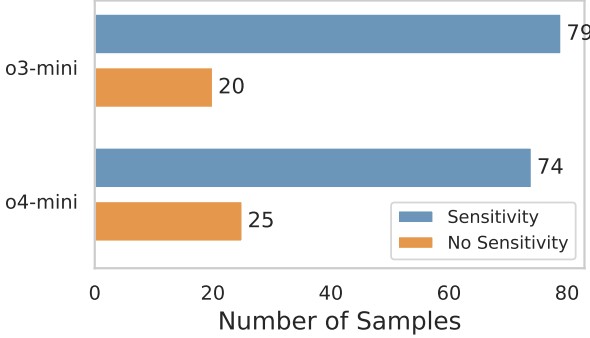

Figure 12: Sensitivity in the SYNTAXVERIFICATION task for reasoning models `o3-mini` and `o4-mini`. Each bar represents frequency of samples where the model was either sensitivity or not to syntactic changes, under two prompts, $\alpha$ and $\beta$ which differ by a small number of token that change the expected output. **Blue** bars represent the *sensitive* cases where the model's output changed in response to the syntactic errors, as expected. **Orange** bars represent *non-sensitive* cases where the model failed to adapt its prediction, despite the change in input.

## B.7 Effect of Different Models for Periodic Generation

We extend our analysis of periodic pattern generation, evaluating how various decoder-only language models perform when tasked with completing structured, periodic sequences $\beta_p^r = (0^{p-1}1)^r 0$. This prefix is defined as $r = 10$ repetitions of a base pattern $0^{p-1}1$ plus a token zero.

Figure 13 presents results for several CPE transformer models of varying sizes. We observe that each model exhibits a *critical period*—a threshold value of $p$ beyond which the model fails to reliably complete the pattern. For smaller periods (e.g., $p \leq 10$), models achieve perfect or near-perfect extrapolation, while performance degrades as the period increases, eventually resulting in complete failure for periods beyond the model-specific threshold. Notably, all models show this limitation except some of them: for examples `gemma-2b` and `gemma-2-2b`.

## C Prompt formatting

We present the prompt format used for test sensitivity in models for SYNTAXVERIFICATION task. Here we follow the same intruction structure for all model adapting this to each specific special tokens. For reasoning model, we consider the same structure in order to ensure comparable results.

```
gemma-3-it

<bos><start_of_turn>user
You are a Python expert. Carefully read the following python codes and answer to the
questions.<end_of_turn>
<start_of_turn>model
OK.<end_of_turn>
<start_of_turn>user
Instruct: Read the following python code and answer the question.
```python
{{python_shot_function1}}
```
Question: Does the Python code compile without syntax errors? If no error is detected, return 1; other
wise, return 0.<end_of_turn>
<start_of_turn>model
Instruct: Read the following python code and answer the question.
Answer: 1<end_of_turn>
<start_of_turn>user
```python
{{python_shot_function2}}
```
Question: Does the Python code compile without syntax errors? If no error is detected, return 1; other
wise, return 0.<end_of_turn>
<start_of_turn>model
Answer: 0<end_of_turn>
<start_of_turn>user
Instruct: Read the following python code and answer the question.
```python
{{python_test_function}}
```
Question: Does the Python code compile without syntax errors? If no error is detected, return 1; other
wise, return 0.<end_of_turn>
<start_of_turn>model
Answer:
```

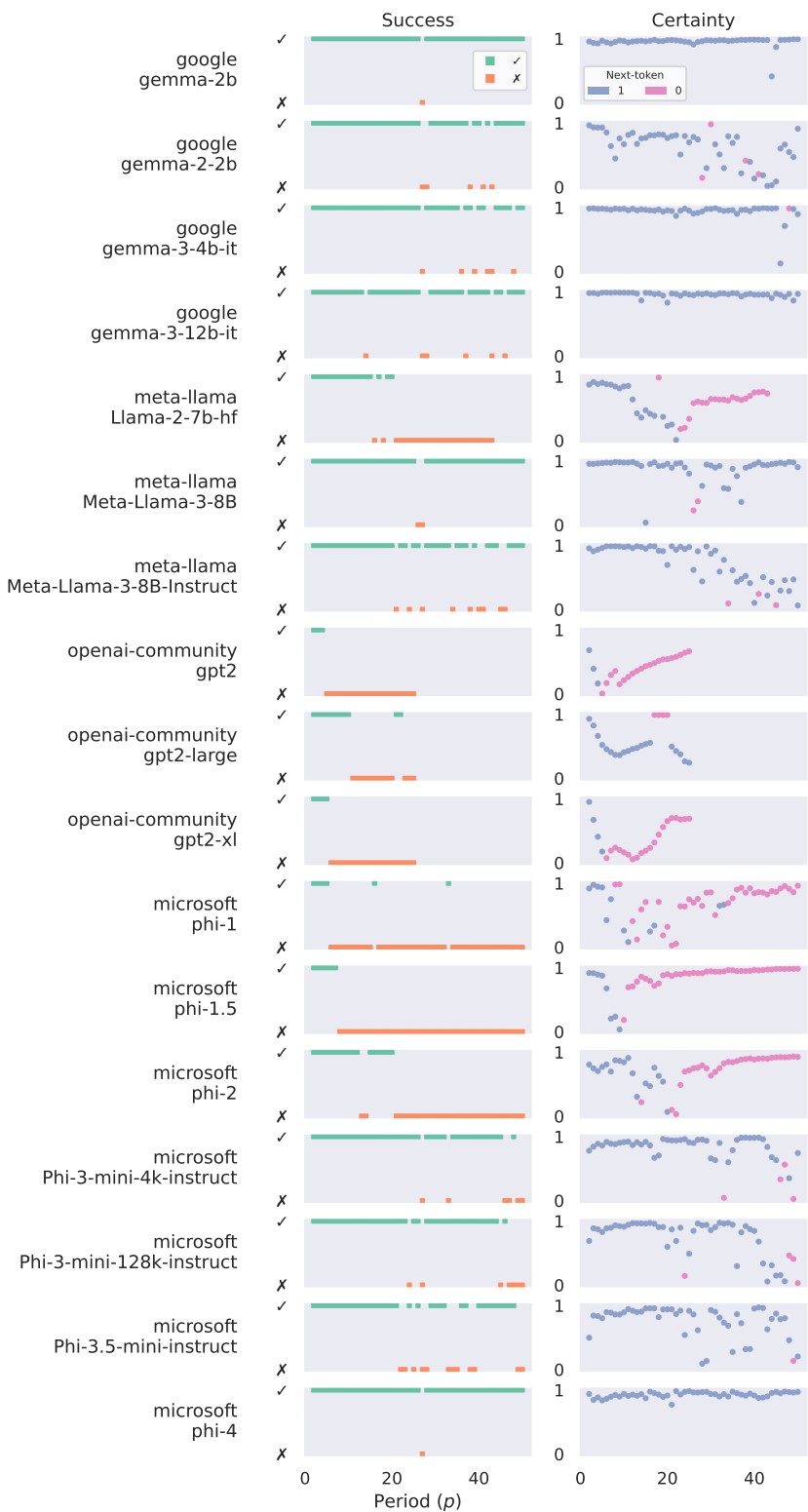

Figure 13: Evaluation of periodic sequence generation using diverse models. **Rows** correspond to fix language model with $r = 10$ repetitions.

```
phi-4

<|im_start|>system<|im_sep|>
You are a Python expert. Carefully read the following python codes and answer to the questions.<|im_end|>
<|im_start|>user<|im_sep|>
Instruct: Read the following python code and answer the question.
```python
{{python_shot_function1}}
```
Question: Does the Python code compile without syntax errors? If no error is detected, return 1; other
wise, return 0.<|im_end|>
<|im_start|>assistant<|im_sep|>
Answer: 1<|im_end|>
<|im_start|>user<|im_sep|>
Instruct: Read the following python code and answer the question.
```python
{{python_shot_function2}}
```
Question: Does the Python code compile without syntax errors? If no error is detected, return 1; other
wise, return 0.<|im_end|>
<|im_start|>assistant<|im_sep|>
Answer: 0<|im_end|>
<|im_start|>user<|im_sep|>
Instruct: Read the following python code and answer the question.
```python
{{python_test_function}}
```
Question: Does the Python code compile without syntax errors? If no error is detected, return 1; other
wise, return 0.<|im_end|>
<|im_start|>assistant<|im_sep|>
Answer:
```

```
Meta-Llama-3-Instruct

<|begin_of_text|><|start_header_id|>system<|end_header_id|>

You are a Python expert. Read the following instructions carefully and respond to the
questions.<|eot_id|><|start_header_id|>user<|end_header_id|>

Instruct: Read the following python code and answer the question.
{{python_shot_function1}}
Question: Does the Python code compile without syntax errors? If no error is detected, return 1; other
wise, return 0.<|eot_id|><|start_header_id|>assistant<|end_header_id|>

Answer: 1<|start_header_id|>user<|end_header_id|>

Instruct: Read the following python code and answer the question.
{{python_shot_function2}}
Question: Does the Python code compile without syntax errors? If no error is detected, return 1; other
wise, return 0.<|eot_id|><|start_header_id|>assistant<|end_header_id|>

Answer: 0<|eot_id|><|start_header_id|>user<|end_header_id|>

Instruct: Read the following python code and answer the question.
{{python_test_function}}
Question: Does the Python code compile without syntax errors? If no error is detected, return 1; other
wise, return 0.<|eot_id|><|start_header_id|>assistant<|end_header_id|>

Answer:
```

