# OpenReview forum: "Continuity and Isolation Lead to Doubts or Dilemmas in Large Language Models"
_NeurIPS.cc/2025/Conference — NeurIPS 2025 poster_

### Official Review · Reviewer_RSLx · 2025-06-23

**Clarity:** 3
**Significance:** 3
**Originality:** 3
**Rating:** 4
**Confidence:** 3

**Summary:**

This paper derives and demonstrates two phenomena that a decoder-only transformer with compact positional encoding exhibits. The first is continuity: given two similar tokenized sequences, the next token prediction is similar for one sequence as for the other sequence, if the sequences have a small enough Hamming distance. Their theoretical claim was examined by looking at LM’s sensitivity to small changes in the prompt, one on zero fundamental sequence and the other on a syntax verification task, in which the model needs to say whether the syntax is correct or not. They observed that most models do not change their predictions when there is a small change in syntax, supporting the continuity argument. The second is isolation, which is about no such transformer can learn all sequences of infinite length, for example, all periodic sequences, because they have a relative hamming distance of 0.  In practice, they demonstrate the effect of isolation on sequences with a small number of differing samples, supporting that decoder-only transformers with compact positional encoding learn a limited set of periodic sequences.

**Questions:**

Line 89, what is CPE transformer?

What is the accuracy of the model in the syntax prediction error detection task?

Is the model sensitivity dependent on sequence length?

I guess figure 2’s emphasis is on the left figure, do we know in closer detail, the message is if there is a small number/proportion of differing signals, then the prediction results are insensitive, and the right plot suggests that the prediction results are sensitive when there is a substantial differing portion of signals, but you cannot see very well from the left side of figure 2. (How come there are still some models that are insensitive to a big portion of differing signals?)

How does the sensitivity result depend on the sequence length? I know it is 190 in the experiment, but what happens for sequence length in the common token range?

What is the accuracy of the model in the syntax prediction error detection task?

**Ethical Concerns:**

["NO or VERY MINOR ethics concerns only"]

**Final Justification:**

This paper gives clear and original mathematical proofs to show two tendencies and limitations for CPE transformers. Although experiments can be made more related to real-world applicable cases, the idea is original, well presented, and has important implications.

**Limitations:**

Yes, limitations are sufficiently discussed in section 5 of the paper.

**Paper Formatting Concerns:**

I did not notice any formatting concerns.

**Quality:**

3

**Strengths And Weaknesses:**

__Strengths:__

 I appreciated the mathematical rigor of the paper, and the theoretical claims are well-defined. The idea is original and connects analysis with understanding transformer behavior. There is a nice connection between the theoretical derivations and practical experiments.

__Weaknesses:__

 My issue with this theoretical idea is that it is too much of a corner case. The authors claim that the lack of these abilities “*severely* constrains their (even mild) intelligent behavior”, which is a bit unsubstantiated. It is also bizarre to assume intelligence depends on the learnability of infinite sequences. I think the evaluation examinations are too tangential to their actual use cases.  Pragmatically, few give infinite sequences to transformers, but just missing this aspect of behaving properly with respect to infinite or very long sequences does not disqualify a transformer from being intelligent. The paper lacks an argument or a demonstration on why this can be a concerning characteristic of the transformer.

The title is not appropriate to its content, as it is unclear what these doubts and dilemmas refer to, the wording deviates from the main findings of the experiments.

Something is missing between the introduction and the contributions. I suggest putting related work before the contribution.

Line 114, 115 on Page 3 is broken.

321-322 is missing a comma

section 1.3’ s title does not match its content

---

> ### Author Rebuttal · Authors · 2025-07-29
>
> We thank the reviewer for carefully reviewing our work and the comments raised. Below, we provide answers to the weaknesses and questions posed by the reviewer.
>
> Weaknesses rebuttals:
>
> * _(Weakness 1)  My issue with this theoretical idea is that it is too much of a corner case. The authors claim that the lack of these abilities “severely constrains their (even mild) intelligent behavior”, which is a bit unsubstantiated. It is also bizarre to assume intelligence depends on the learnability of infinite sequences. I think the evaluation examinations are too tangential to their actual use cases. Pragmatically, few give infinite sequences to transformers, but just missing this aspect of behaving properly with respect to infinite or very long sequences does not disqualify a transformer from being intelligent. The paper lacks an argument or a demonstration on why this can be a concerning characteristic of the transformer._
>
>
> We note that Theorem 1 (our main result) is actually formulated for finite sequences. The conclusions we derive from it for infinite sequences are meant to illustrate a concrete limitation to perform inductive reasoning, understood as the ability to derive general rules and principles from the presented information (an ability usually considered as a key aspect of intelligent behavior). The limitation for infinite sequences means that the transformer will never be able to understand the rule that generates the sequence, regardless of how long the prompt is. To substantiate even more our claims, we have performed an additional experiment on a novel natural language task, analogous to the Code Syntax Verification from our paper.
>
> The new dataset consists of 56 examples derived from the "babi-nli" dataset. Each instance includes a premise, a hypothesis, and a label (0 or 1). To meet theoretical requirements, we lengthen the premise using neutral text from Wikipedia, creating a "Long-Premise" that preserves the original label and constitutes the alpha-sequence. To create the corresponding "beta-sequence" we modified a single key word at the beginning making sure that it entails a change to the original label.
>
> For example,
> Premise Alpha (Label=True): Yann is hungry. Jason is bored. Antoine is hungry. Yann went back to the kitchen. Yann picked up the apple there....
>
> Premise Beta (Label=False):  Yann is hungry. Jason is bored. Antoine is hungry. Yann went back to the kitchen. Yann picked up the pear there....
>
> Hypothesis: Yann picked the apple because she was hungry.
>
> Considering that Meta-Llama-3-8B-Instruct provided the best results in the Code Syntax Verification task, we tested this model on the new NLI task in two scenarios with different input lengths. Namely ~1000 tokens and ~5000 tokens. The results were as follows:
>
> Sensitivity:   25% (about 1000 tokens),   14% (about 5000 tokens)
> Accuracy:    62%  (about 1000 tokens),   57% (about 5000 tokens)
>
> As we can see from the results, even in this simple natural language inference task, the sensitivity remains fairly low. We acknowledge that this is an important practical application of Theorem 1. Sensitivity to small changes in a long text is critical for some real-life applications like analyzing legal documents.
>
> * _(Weakness 2) The title is not appropriate to its content, as it is unclear what these doubts and dilemmas refer to, the wording deviates from the main findings of the experiments._
>
> The term “doubts“ refers to the lack of certainty for the next token prediction, while “dilemmas” means that if two prompts are too similar then the transformer can make the right prediction with certainty for at most one of them (in a sense, “it has to choose”). As explained in the paper and tested in the experiments, these two features come from the continuity phenomenon that we study. This is discussed in Section 5 (Conclusion - Doubts and Dilemmas). The referee is correct in that we should explain this earlier in the paper, and we will do so in the CR-version.
>
>
> * _(Question 1) “Line 89, what is CPE transformer?”_
>
> CPE is first introduced in line 43 and stands for compact positional encoding. A formal definition of CPE is later given in line 174. The CR-version will formally define CPE when it first appears in the text.
>
> * _(Question 2) What is the accuracy of the model in the syntax prediction error detection task?_
>
> The accuracies for the different models are 52% (gemma-3-4b-it), 54% (gemma-3-12b-it), 49% (phi-4) and 60% (Meta-Llama-3-8B-Instruct). We will add this information to the CR-version.
>
> * _(Question 3) Is the model sensitivity dependent on sequence length?_
>
> Theorem 1 imposes a universal relationship between epsilon and delta that does not depend on the length of the sequence. However, if for instance we are perturbing a single token (as in the syntax experiment), then the Normalized Hamming distance between the two prompts is 1/n where n is their length. Thus, for larger prompts the resulting perturbation is smaller, and therefore more likely to show non-sensitivity.  We have in fact tested this. In Appendix B.5, Figure 10, we present the results of the Code Syntax Verification task on a series of other models using prompts with around 300 tokens (as opposed to the results in the main body of the paper which stemmed from inputs of more than 1000 tokens) and observe improved sensitivity.
>
> * _(Question 4) I guess figure 2’s emphasis is on the left figure, do we know in closer detail, the message is if there is a small number/proportion of differing signals, then the prediction results are insensitive, and the right plot suggests that the prediction results are sensitive when there is a substantial differing portion of signals, but you cannot see very well from the left side of figure 2. (How come there are still some models that are insensitive to a big portion of differing signals?)_
>
> Observing insensitivity even for large perturbations is permitted by Theorem 1, although not guaranteed in all cases. Our results guarantee insensitivity for sufficiently small perturbations.
>
> * _(Question 5) “How does the sensitivity result depend on the sequence length? I know it is 190 in the experiment, but what happens for sequence length in the common token range?”_
>
> Our theoretical results predict that non-sensitivity is more likely to appear with longer sequences. Indeed, as illustrated in Appendix B.5 (Figure 10), on average we observe an improvement in sensitivity when the sequence length is shorter. Moreover, the novel natural language task described in weakness 1 also reveals this phenomenon.

---

> > ### Comment · Reviewer_RSLx · 2025-08-03
> >
> > Thank you to the authors for their detailed and thoughtful rebuttal. I appreciate the additional experiments and clarifications provided; the paper illustrates an important aspect of CPE transformers with practical implications. I would therefore raise my score to 4.

---

> > > ### Author Response · Authors · 2025-08-05
> > >
> > > We thank again the reviewer for the time invested in raising the questions and comments that improved the quality and scope of our work. Following the reviewer's suggestion to raise the score to 4, we kindly ask if she/he can validate this change within the platform before the deadline.

---

### Official Review · Reviewer_DY31 · 2025-06-27

**Clarity:** 4
**Significance:** 2
**Originality:** 3
**Rating:** 5
**Confidence:** 4

**Summary:**

The this paper really on two simple (yet non-trivial) observations:
- Continuity: a Transformer outputs can not change too much when its prompt does not change much
- Isolation: as a consequence, a Transformer can not learn simultaneously infinite sequences that are equal on a dense set, sooner or later it will have to make next token prediction error on one of the two sequences.

**Questions:**

It has been shown that hybrid models between Mamba and Transformers work better than Transformers for long context (https://github.com/NVIDIA/RULER). Do you have any intuition of what could explain these observations? Would your work allow you to shed some lights on that matter? (Maybe Mamba models suffers a representation collapse at a lower speed)

Does your study make you bullish on approaches such as https://arxiv.org/pdf/2501.19399?

**Ethical Concerns:**

["NO or VERY MINOR ethics concerns only"]

**Final Justification:**

I liked the paper overall. However, the authors’ answers to my questions seem relatively superficial, as I suspect some form of representational collapse in the state-space model, and the authors did not provide strong evidence that their theory explains the success of Jamba in practice. Nevertheless, in my view, the paper is an "accept."

**Limitations:**

Yes

**Quality:**

3

**Strengths And Weaknesses:**

**Strength**:
- In essence, the paper shows that Transformers sees infinite sequences from the Besicovitch space (which is much smaller than the Weyl or Cantor spaces).
- The results do not see too toyish, but of practical relevance at scales in which Transformers operate in practice.
- It could explain the difficulty to do long-context with transformers, and justify the use of CoT, memory layers, tools...
- The paper is really well written.

**Weaknesses**:
- It would be nice to have a finer characterization of the modulus of continuity in Theorem 1. In particular, I assume that the MLP layers F could introduce big variations that may make this work irrelevant in practice (e.g. I won't be surprised if someone show that one can modify a single token in a prompt to lead to completely different outputs for any decently long sequences, and that the conflict with this paper would be due to the modulus of continuity of the MLP layers).
- It may be the case that the results are brittle, and do not really explain as many practical observations on long context as one may want to believe. In particular, I am not sure that the theoretical tools used in this paper could explain the observations made on the RULER benchmark (see question below).

---

> ### Author Rebuttal · Authors · 2025-07-29
>
> We thank the reviewer for the time invested in carefully reviewing our work and the valuable comments raised during this process. Below, we provide answers to the questions posed by the reviewer.
>
> * _(Question 1) It has been shown that hybrid models between Mamba and Transformers work better than Transformers for long context (https://github.com/NVIDIA/RULER). Do you have any intuition of what could explain these observations? Would your work allow you to shed some lights on that matter? (Maybe Mamba models suffers a representation collapse at a lower speed)._
>
> As Merrill [1] has shown,  simple state space models as RNNs can recognize any regular language. In particular, there exists a simple RNN which on any long sequence input  like “0000…0” outputs 0 but on a sequence like “1000…0” outputs 1. At the same time, by Theorem 1, Transformers with compact positional encoding (such as RoPE) cannot do that.  Theorem 1 can therefore be seen as a separation result between the capabilities of these two architectures. In turn, this suggests an explanation why incorporating SSM modules into a transformer architecture (as in Jamba) may in principle result in models overcoming representational collapse.
>
> * _(Question 2) Does your study make you bullish on approaches such as https://arxiv.org/pdf/2501.19399?_
>
> This is also an interesting point. We have in fact analysed it in Section B.4 of the Appendix where we tested the effect of replacing softmax in the attention layers by the so-called scalable softmax, which is a way to incorporate non-compact components. The resulting model therefore escapes from the framework of our theory, and is thus not subordinated to the constraints imposed by our results. And indeed, as can be seen from Figure 8, our experiment shows a significant improvement in sensitivity. We acknowledge that this observation is absent from the main body of the paper. We will correct this in the revised version.

---

> > ### Comment · Reviewer_DY31 · 2025-08-04
> > **Thank you**
> >
> > Thank you for your answer.
> > I wonder how much the provided example where the RNN flags the existence of a pattern really far in the past (the 1 in the sequence of 100000…) is relevant to explain the success of Jamba. However this is clearly beyond this rebuttal.

---

### Official Review · Reviewer_taZZ · 2025-06-30

**Clarity:** 3
**Significance:** 4
**Originality:** 3
**Rating:** 5
**Confidence:** 4

**Summary:**

The core of this paper is to deepen the understanding of the mechanism of Transformer by conceptualizing and analyzing two phenomenon:

1)	Isolation expresses that any learnable sequence must be isolated from another learnable sequence, and hence some sequences cannot be learned by a single Transformer at the same time.

2)	Continuity entails that an attractor basin forms around a learned sequence, such that any sequence falling in that basin will collapse towards the learned sequence.

The authors provide not only the mathematic proof, but also the careful designed experiment to verify the claim.

**Questions:**

Can similar findings be applied to RNNs? Additionally, could you suggest possible architectural designs that might address the limitations identified in this paper?

**Ethical Concerns:**

["NO or VERY MINOR ethics concerns only"]

**Final Justification:**

1) The response to Question 1 was particularly insightful. I had previously misunderstood the limitations discussed as being common to all neural networks. However, the clarification that these limitations are specific to the Transformer architecture was very helpful.

2) In addition, the new experimental results are quite compelling. The observed relationship between the number of tokens and sensitivity is especially interesting.

These two points have led me to reassess the contribution of this paper. I am now convinced that it offers valuable insights for the future development of neural network architectures.

**Limitations:**

See weakness

**Quality:**

3

**Strengths And Weaknesses:**

Strength:

1.  Overall, this paper is well written.
2.  The paper offers a valuable perspective on analyzing the Transformer architecture.

Weakness:
1.  The paper appears to be poorly positioned. While the title suggests a focus on large language models (LLMs), the main content primarily discusses Transformers. Although LLMs are typically built on Transformer architectures, the two are not synonymous, and this distinction is not clearly addressed.
2. Several prior works [1,2,3] have explored the fundamental limitations of the general Transformer architecture, but this paper does not sufficiently engage with or contextualize those contributions.
3.  The findings of this paper are limited to pattern discovery in purely numerical sequences. It remains unclear whether these results can be generalized to natural language. In most cases, natural language understanding does not rely on the kinds of patterns examined in the paper. Moreover, natural language carries significantly richer semantic content than the simplified tasks considered here.

[1] Faith and Fate: Limits of Transformers on Compositionality

[2] Limits of Transformer Language Models on Learning to Compose Algorithms

[3] On Limitations of the Transformer Architecture

---

> ### Author Rebuttal · Authors · 2025-07-29
>
> We thank the reviewer for the careful read of our manuscript and the valuable comments raised. The reviewer is concerned about a better positioning of our work with respect to previous work, but also the scope of our findings. Below, we provide a point-by-point reply to the weaknesses and questions raised by the reviewer.
>
>
> * _(Weakness 1) The paper appears to be poorly positioned. While the title suggests a focus on large language models (LLMs), the main content primarily discusses Transformers. Although LLMs are typically built on Transformer architectures, the two are not synonymous, and this distinction is not clearly addressed._
>
>
> We thank the reviewer for this remark. Our camera-ready version will position the scope of our paper taking into account this comment, and we will make sure to isolate our claims to transformer-based LLMs from the introduction onwards.
>
> * _(Weakness 2) Several prior works [1,2,3] have explored the fundamental limitations of the general Transformer architecture, but this paper does not sufficiently engage with or contextualize those contributions._
>
> We thank the reviewer for pointing out these references. To improve the positioning of the paper, we will add the discussion of these references in the related work section.
>
>
> * _(Weakness 3) The findings of this paper are limited to pattern discovery in purely numerical sequences. It remains unclear whether these results can be generalized to natural language. In most cases, natural language understanding does not rely on the kinds of patterns examined in the paper. Moreover, natural language carries significantly richer semantic content than the simplified tasks considered here._
>
> We note that Theorem 1 (our main result) is agnostic to the nature of the input and thus applies equally well to natural language or any other kind of input. The consequences we present for pattern recognition over numerical sequences are only meant as a concrete illustration of the limitations implied by Theorem 1 on inductive reasoning, understood as the ability to derive general rules and principles from the presented information (and usually considered as a key aspect for human intelligence). We also note that besides numerical sequences,  section 3.2 of our paper presents experiments on a syntax verification task that requires both advanced semantic and complex syntactic structure understanding. Nonetheless, to address the concern raised by the reviewer, we have performed an additional experiment on a novel natural language task, analogous to the Code Syntax Verification task.
>
> The new dataset consists of 56 examples derived from the "babi-nli" dataset. Each instance includes a premise, a hypothesis, and a label (0 or 1). To meet theoretical requirements, we lengthen the premise using neutral text from Wikipedia, creating a "Long-Premise" that preserves the original label and constitutes the alpha-sequence. To create the corresponding "beta-sequence" we modified a single key word at the beginning making sure that it entails a change to the original label.
>
> For example,
> Premise Alpha (Label=True): Yann is hungry. Jason is bored. Antoine is hungry. Yann went back to the kitchen. Yann picked up the apple there....
>
> Premise Beta (Label=False):  Yann is hungry. Jason is bored. Antoine is hungry. Yann went back to the kitchen. Yann picked up the pear there....
>
> Hypothesis: Yann picked the apple because she was hungry.
>
> Considering that Meta-Llama-3-8B-Instruct provided the best results in the Code Syntax Verification task, we tested this model on the new NLI task in two scenarios with different input lengths. Namely ~1000 tokens and ~5000 tokens. The results were as follows:
>
> Sensitivity:   25% (about 1000 tokens),   14% (about 5000 tokens)
>
> Accuracy:    62%  (about 1000 tokens),   57% (about 5000 tokens)
>
> As we can see from the results, even in this simple natural language inference task, the sensitivity remains fairly low. We acknowledge that this is an important practical application of Theorem 1. Sensitivity to small changes in a long text is critical for some real-life applications like analyzing legal documents.
>
>
> * _(Question 1) Can similar findings be applied to RNNs? Additionally, could you suggest possible architectural designs that might address the limitations identified in this paper?_
>
> This is an interesting question. The Continuity phenomenon does not apply to RNNs. Indeed, Merrill [1] has shown that simple RNNs can recognize any regular language. In particular, there exists a simple RNN which on any long sequence input  like “0000…0” outputs 0 but on a sequence like “1000…0” outputs 1. At the same time, by Theorem 1, Transformers with compact positional encoding (such as RoPE) cannot do that.
>
>  Theorem 1 can therefore be seen as a separation result between the capabilities of these two architectures. In turn, this suggests that suitably incorporating state space models modules into a transformer architecture (as in Jamba) may help to overcome the limitations imposed by continuity and isolation. Another possibility to overcome these limitations is to allow non-compact components. This is in fact illustrated in Appendix B.4 of our paper, where we tested the effect of replacing softmax in the attention layers by the so-called scalable softmax, which is a version of softmax that explicitly incorporates a term that grows logarithmically with the size of the input, therefore introducing non-compactness. As can be seen from Figure 8, our results unequivocally show that replacing standard softmax by scalable softmax (while leaving everything else unchanged) leads to a significant improvement in sensitivity. Finally, another possibility to overcome continuity limitations is to allow non-compact positional encodings.
>
> [1] William Merrill. 2019. Sequential neural networks as automata.

---

> > ### Comment · Reviewer_taZZ · 2025-08-06
> >
> > Thank you to the authors for the detailed response.
> >
> > 1) The response to Question 1 was particularly insightful. I had previously misunderstood the limitations discussed as being common to all neural networks. However, the clarification that these limitations are specific to the Transformer architecture was very helpful.
> >
> > 2) In addition, the new experimental results are quite compelling. The observed relationship between the number of tokens and sensitivity is especially interesting.
> >
> > In light of these points, I intend to raise my score from 3 to 5.

---

### Official Review · Reviewer_yoGw · 2025-07-02

**Clarity:** 3
**Significance:** 3
**Originality:** 3
**Rating:** 5
**Confidence:** 3

**Summary:**

This papers provides two interesting fundamental limitations in the learning capability of transformers, namely continuity and isolation. Continuity shows that prompts the are close enough always produces probability output that are also close. On the other hand, isolation shows that for any two infinitely long sequences that are learnable by a transformer, if the two sequences differ at infinitely many places, then they must be far enough in Hamming distance. The paper also shows experimental analysis showing the importance of the limits in learning in several synthetic as well as some practical scenarios.

**Questions:**

Please also see the weaknesses.
Additional questions:
Line 223: "Middle positions result in lower sensitivity, and early tokens yield almost no sensitivity at all"
Q: does the theory explain this observation?

Line 196: "same last token"
Why is this condition in theorem 1 important, please explain intuitively.

**Ethical Concerns:**

["NO or VERY MINOR ethics concerns only"]

**Final Justification:**

I thank the author for their detailed response with clarifications and the additional experiment.

It clarifies the concerns/questions I had regarding infinite sequence: I went through the proof again and I see that it is indeed for finite sequence. I was confused with the infinite sequence examples in the paper. I would suggest having some finite sequence example as well in the final version of the work to avoid any confusion for readers.

Thanks for clarification on the figures, I would suggest making the captions more detailed so that it is easy for the readers to understand the figure.

Thanks for the experiments on the reasoning models. It is indeed interesting that reasoning does not help much with the continuity issue.
I have increased the score to reflect the above.

**Limitations:**

yes

**Paper Formatting Concerns:**

No concerns

**Quality:**

3

**Strengths And Weaknesses:**

Strengths
- The paper theoretically illustrates two fundamental limitations in the learning capability of transformers

- The paper also shows experiments illustrating the continuity and isolation phenomenon on synthetic tasks such as number sequence generation as well as practical applications such as syntax verification tasks.

- The paper is well written in terms of the clarity of the results and simple experiments supporting and illustrating the claims.

Weaknesses:
- The theorems are all claims for infinite sequences. Hence, it is difficult to judge how applicable they are practically beyond the illustrated synthetic examples. Is it possible to show more convincing practical usecases such as the code-syntax related.

- For Fig. 3 the pattern does not seem to follow same as in Fig. 2 for continuity. Please explain if I am misunderstanding the figures.

- As also mentioned in the limitations, the analysis lacks consideration of chain-of-thought reasoning generated by recent reasoning models. Is it possible that these reasoning traces allow different output after reasoning, hence, not be limited by the results in the theorem. Can we check it empirically on some of the synthetic datasets using some reasoning models?

---

> ### Author Rebuttal · Authors · 2025-07-29
>
> We thank the reviewer for the positive appraisal of our work and greatly appreciate the time invested in reviewing it. Below, we reply to the three weaknesses raised by the reviewer, followed by replies to the questions raised.
>
> * _(Weakness 1) The theorems are all claims for infinite sequences. Hence, it is difficult to judge how applicable they are practically beyond the illustrated synthetic examples. Is it possible to show more convincing practical use cases such as the code-syntax related._
>
> We note that Theorem 1 (our main result) is in fact formulated for finite sequences. The conclusions for infinite sequences are presented only to illustrate clear limitations on a task requiring inductive reasoning, namely, the ability to derive general rules and principles from the presented information (an ability considered as a key aspect of intelligence behavior). In addition to the code syntax verification task, and to address the reviewers comment, we have tested the corollary of Theorem 1 on a new natural language task.
>
> We evaluate LLMs on a natural language inference (NLI) task analogous to the Code Syntax Verification task. The new dataset consists of 56 examples derived from the "babi-nli" dataset. Each instance includes a premise, a hypothesis, and a label (0 or 1). To meet theoretical requirements, we lengthen the premise using neutral text from Wikipedia, creating a "Long-Premise" that preserves the original label and constitutes the alpha-sequence. To create the corresponding "beta-sequence" we modified a single key word at the beginning making sure that it entails a change to the original label.
>
> For example,
> Premise Alpha (Label=True): Yann is hungry. Jason is bored. Antoine is hungry. Yann went back to the kitchen. Yann picked up the apple there....
>
> Premise Beta (Label=False):  Yann is hungry. Jason is bored. Antoine is hungry. Yann went back to the kitchen. Yann picked up the pear there....
>
> Hypothesis: Yann picked the apple because she was hungry.
>
> Considering that Meta-Llama-3-8B-Instruct provided the best results in the Code Syntax Verification task, we tested this model on the new NLI task in two scenarios with different input lengths. Namely ~1000 tokens and ~5000 tokens. The results were as follows:
>
> Sensitivity:   25% (about 1000 tokens),   14% (about 5000 tokens)
>
> Accuracy:        62% (about 1000 tokens),   57% (about 5000 tokens)
>
> As we can see from the results, even in this simple natural language inference task, the sensitivity remains fairly low. We acknowledge that this is an important practical application of Theorem 1. If the reviewer agrees we can add these findings in the revised version.
>
> * _(Weakness 2) For Fig. 3 the pattern does not seem to follow same as in Fig. 2 for continuity. Please explain if I am misunderstanding the figures._
>
> Figures 2 and 3 are plots of different things, it is therefore normal that the patterns are different. In Fig. 2 the x-axis corresponds to the percentage of tokens that were randomly changed in order to produce the perturbed sequence, while in Figure 3 this perturbation size is fixed. The purpose of Fig. 2 is to show the number of times (out of 100) a given perturbation size produced a change in the output. In contrast, in Fig. 3 the plot is meant to compare the output probability of the most likely token given the original sequence as input (alpha)  against the output probability of the same token, but given instead the perturbed sequence (beta).
>
> * _(Weakness 3) As also mentioned in the limitations, the analysis lacks consideration of chain-of-thought reasoning generated by recent reasoning models. Is it possible that these reasoning traces allow different output after reasoning, hence, not be limited by the results in the theorem. Can we check it empirically on some of the synthetic datasets using some reasoning models?_
>
> This is an interesting point, which we in fact discuss in Section B.6 of the Appendix. In particular, we tested Open AI’s o4-mini and o3-mini models on the syntax verification task. Strikingly, we observe that even on our very simple task, these models still display a level of error greater than 20%. This suggests that while CoT may slow down the effect of continuity, it may not completely eliminate it. We acknowledge that these results are omitted from the main body of the paper. We will make sure to correct this in the revised version.
>
> *  _(Question 1) Line 223: "Middle positions result in lower sensitivity, and early tokens yield almost no sensitivity at all" Q: does the theory explain this observation?_
>
> No, our theoretical results do not account for differences in sensitivity associated with the positions of the perturbations (beginning, middle or end).
>
> * _(Question 2) Line 196: "same last token" Why is this condition in theorem 1 important, please explain intuitively._
>
> Intuitively, because one can build a transformer that always outputs the last token (ignoring previous ones).
> For that transformer, changing the last token changes the output regardless of the input length.
> Hence, the condition about the last token is strictly necessary for the theorem 1.

---

### Decision · Program_Chairs · 2025-09-17

**Decision:**

Accept (poster)

**Comment:**

This article considers the ability of transformers that use compact positional encodings to distinguish strings of tokens from each other with ‘certainty’ defined as a guaranteed margin for correctly predicting the continuations of distinct strings.  A mathematical theory of novel termology is introduced, and two theorems and their informal corollaries are presented, with proofs in a supplement.  The key claim of the article, stated informally/approximately, is that as strings get longer distinctions can become harder to detect with certainty.   Conceptual examples center on easy to understand sequences like sequences of all 0’s or simple periodic sequences, while several practical examples are also considered, showing that all transformers tested have limits in their ability to produce valid responses for some long sequences while correctly producing distinct responses for other valid sequences.

After reviews and rebuttals, the paper received ratings of 5,5,5, and 4 – that is 3 of four reviewers recommended acceptance while one rated the paper a borderline accept.  The strength of the paper, as the reviewers saw it, was that it clearly delineated in-principle limits on compact transformers potential, while also showing that these limits do show up in practice.   Several scores were initially lower but through the rebuttal process several reviewers came to understand the paper better, and a new practical example using natural language was introduced, addressing most of the weaknesses identified.

I am pleased to join the reviewers in recommending acceptance – it is clear that many members of the NeurIPS community will consider themselves to have been informed or to have had their understanding of phenomena they are dimly aware of clarified by the paper and its findings.   I do hope that the authors will revise the paper to ensure that some of the initial misunderstandings expressed by individual reviewers are addressed, and to include the new natural language task results, which one of the reviewers found illuminating.

I would, however, like to urge the authors to consider tempering some of the stronger statements in the paper and slightly adjusting their framing of the importance of their findings as I discuss below.

The paper argues that we should expect an intelligent system to be capable of processing distinct sequences of arbitrary length.   Certain computational systems may have this capability (under the assumption of infinite memory) but to consider a system intelligent only if it has this capability is not something that I think the field of computational intelligence should accept without discussion.  On this view, humans (without recourse to external supports such as the ability to read and write and or to rely on computer programs) would not be considered intelligent.  Chomsky made the move of accepting capacity limitations as a fact of human nature as part of his defense of a computational formulation of one important aspect of human intelligence.

A related point is that the models under consideration can be quite successful up to fairly long sequence lengths and thus they can be of considerable use in practice, and I think it would be worthwhile for the presentation to more explicitly acknowledge this point. Indeed in my view the paper’s value lies primarily in providing a starting place for considering the limitations of particular types of computational systems. While related points are currently made in the discussion and limitations section, it would strengthen the paper if this point could be more fully acknowledged.  Additionally and if possible, some of the other points that came up in the exchange with reviewers about the pros and cons of alternative architectures including state space machines and RNNs could at least be hinted at within the paper somewhere.